# Learning to Deaggregate: Large-scale Trajectory Generation with Spatial Priors

## Abstract

Generating realistic large-scale trajectories is essential for applications in urban mobility and transportation, yet current generative models either do not offer any controllability or rely on strong sample-specific conditioning. We introduce the Temporal Deaggregation Diffusion Model (TDDM), a hierarchical framework that first represents mobility using spatial priors, which are marginal distributions over geographical occupancy, and then deaggregates them into trajectories. This separation enables generation without sample-specific conditions, supporting transfer to new regions. To support evaluation, we build a benchmark across three cities spanning different continents (Asia, Europe, North America), with standardized metrics for fidelity and distributional coverage. Across all datasets, TDDM improves trajectory fidelity and coverage over leading baselines, and demonstrates stable performance when applied to unseen cities. By explicitly decoupling spatial allocation from temporal realization, our work highlights the role of spatial occupancy priors in enabling scalable and generalizable trajectory generation.

## 1 Introduction

Time-series data of human mobility enables applications such as pandemic forecasting and management (Ilin et al., 2021), smart city development (Wang et al., 2022), urban governance (Xiong et al., 2024), human rights violation detection (Tai et al., 2022) and monitoring of global migration induced by war and climate change (Niva et al., 2023; Alessandrini et al., 2020). Two major challenges stand in the way for using time-series data for these purposes.

The first is a *shortage of publicly available data* (Ansari et al., 2024). Data can only be collected and shared in limited capacity due to concerns of privacy, business and national security, creating a silo effect. Secondly, *generalization beyond observed data*, such as to new regions, unseen spatial areas, or rapidly changing environments, is often a necessary complement to the readily collectible data. One such case is generating high-fidelity realistic spatio-temporal trajectory data, such as individual pedestrians navigating a city or a building. Open problems within the road traffic domain (Lana et al., 2018), and using human mobility data at large, are (1) high quality large-scale trajectories and (2) adaptation to sudden environmental changes. Both are hindered by the unavailability of data, either because existing data cannot be shared or because new environments lack sufficient observations.

A promising direction is to use time-series generative models to capture and generalize mobility distributions. Although these models can be adapted for privacy (Yoon et al., 2019b; Wang et al., 2023; Buchholz et al., 2024) or forecasting (Alcaraz & Strodthoff, 2023), this work focuses exclusively on improving fidelity and cross-region generalization.

Existing approaches, while promising, either fail to capture the multi-modal structure of mobility data or struggle to scale across diverse environments (Buchholz et al., 2024). GAN- and VAE-based approaches such as TimeGAN (Yoon et al., 2019a), TimeVAE (Desai et al., 2021), COSCI-GAN (Seyfi et al., 2022), and TrajGen (Cao & Li, 2021) suffer from mode collapse and oversimplified representations, with unconditional methods additionally offering no control over generated patterns. Recent diffusion models improve fidelity but either remain unconditional (Diffusion-TS (Yuan & Qiao, 2024)) or rely on strong sample-specific conditioning (DiffTraj (Zhu et al., 2023), ControlTraj (Zhu et al., 2024)), limiting generalization across regions. While sample-specific conditioning increases

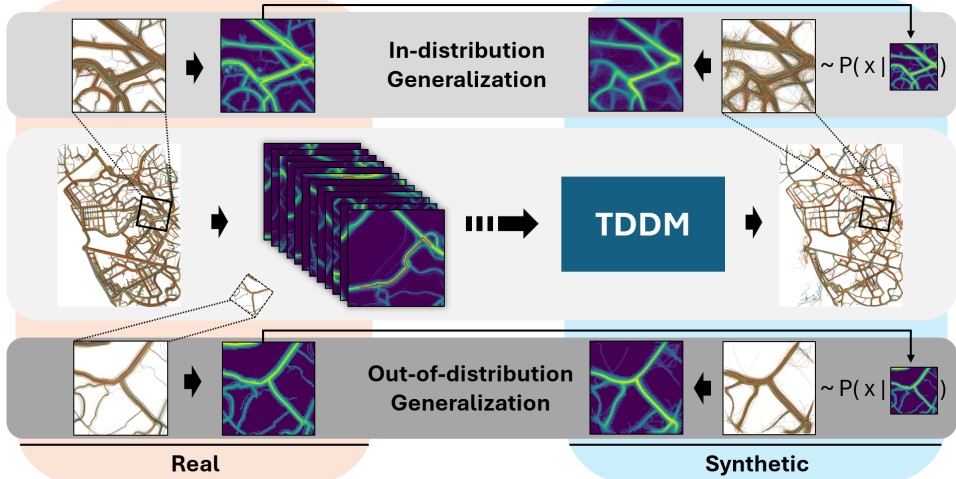

Figure 1: TDDM is trained on real 2D trajectories (left) to generate synthetic trajectories (right), conditioned on how likely it should be for the population of synthetic trajectories to occupy space on the 2D plane (spatial prior). The latter is represented as a discrete distribution over occupancy frequency, i.e., the marginal distribution over the trajectory probability distribution if integrating out time. This yields both high-fidelity in-distribution generalization (top) and out-of-distribution generalization (bottom), the latter when conditioning on a marginal distribution not part of the training data (dashed rectangle). Trajectory data from Beijing, China (Geolife).

quality of the synthetic trajectories, it increases the risk of memorization and prevents cross-region generalization by potentially tying each sample to a training example. Our key insight is that trajectory generation can be factorized into two components: *where* people move, encoded as spatial priors, and *how* people move temporally.

To this end, we introduce Temporal Deaggregation Diffusion Model (TDDM), which conditions on spatial aggregate priors rather than trajectory-level statistics, combining controllability with high fidelity and cross-region generalization. We capture spatial patterns through marginal distributions over local regions, then condition a diffusion model to generate temporal trajectories that respect these patterns. Critically, we canonicalize each region before modeling via a similarity transform, enabling a single model to generalize across all regions for a given scale (e.g. 3x3 km). This spatial-temporal separation enables city-to-city generalization: the model learns temporal dynamics that are invariant to absolute location and orientation. Figure 1 illustrates capabilities of TDDM in a setting of mobility trajectory data. Figure 2 provides a visual comparison with baseline methods.

Our main contributions are:

- *Spatial-Temporal Factorization*: We propose TDDM, a diffusion-based trajectory model that factorizes generation into spatial occupancy priors and temporal dynamics, with coordinate normalization enabling parameter sharing across geographic regions.

- *Benchmarking at Scale*: We establish a standardized evaluation framework across three cities on different continents (Beijing, Porto, San Francisco), with trajectory-specific metrics that harmonize sample fidelity, distributional coverage, and downstream usefulness.

- *Improved Fidelity and Coverage*: TDDM consistently outperforms leading baselines on KL-based distributional measures, demonstrating improved support coverage and proportionality while maintaining strong fidelity across datasets.

- *Generalization to New Regions*: Leveraging spatial priors and canonicalization, TDDM generates realistic trajectories in unseen parts of a city and in entirely new cities without retraining or finetuning, showing strong out-of-distribution zero-shot performance.

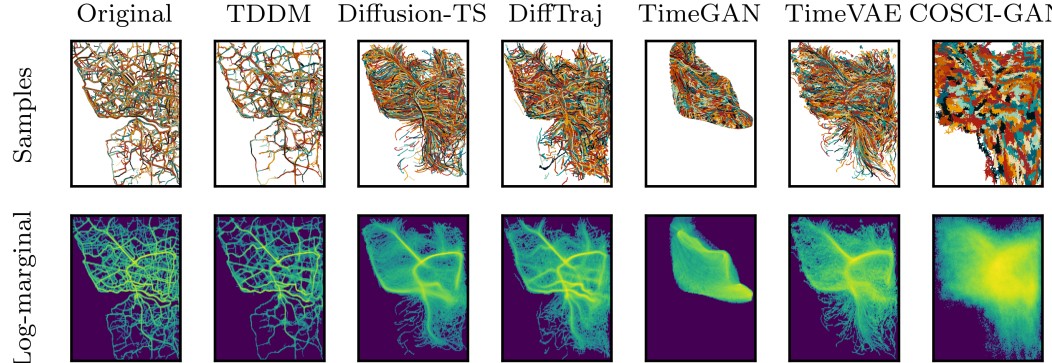

Figure 2: Comparison between original and synthetic trajectories for Porto dataset. First row shows individual trajectory samples while bottom shows log-density heatmaps of all observations. The synthetic data of the proposed model, TDDM, (second column) most closely matches both the individual trajectory patterns and overall density distribution of the original data.

## 2 PROBLEM DEFINITION

The task of learning an unconditional generative model is defined as learning a mapping $f$ from samples drawn from a known distribution $\mathcal{D}_{\text{known}}$, e.g. standard normal distribution, to samples from an unknown target distribution $\mathcal{D}_{\text{unknown}}$. The mapping is learned without direct access to the unknown distribution, and is instead limited to a set of samples $\mathbb{X}_{\text{train}} = \{x_1, \ldots, x_N\}$, where $x_i \sim \mathcal{D}_{\text{unknown}}$. Once the mapping has been learned, synthetic data can be generating by first sampling the known distribution, then passing the individual samples through the mapping function $\mathbb{X}_{\text{synthetic}} = \{f(y_i)\}_{i=1}^{M}$, where $y_i \sim \mathcal{D}_{\text{known}}$. The goal of this mapping is for the synthetic samples to be *similar* to samples from the known distribution, both on a sample level and as a distribution.

## 3 TEMPORAL DEAGGREGATION DIFFUSION MODEL

In this work, we focus on the issue of learning and controlling distribution properties of the synthetic data distribution. More specifically, we focus on controlling the spatial marginal of trajectories, i.e. where time has been marginalized out. This is achieved by conditioning the generative process on a description of the marginal distribution, a spatial prior, with the goal being to generate samples that in aggregate follows said marginal distribution.

The central idea of TDDM is to separate *where* people move from *how* they move in time. We achieve this by conditioning trajectory generation on spatial priors $H$, which describe marginal occupancy distributions over regions. By factoring spatial allocation from temporal dynamics, TDDM learns invariant motion patterns that generalize across cities.

Concretely, the approach proceeds in three steps: (i) partition the spatial domain into regions, (ii) canonicalize each region via a similarity transform, and (iii) compute region-specific spatial priors that describe the local marginal distribution.

**Partitioning the spatial domain.**   We adopt a hierarchical factorization by partitioning the spatial domain into discrete regions, each associated with a spatial prior. The prior describes the marginal distribution of occupancy in that region after marginalizing out time. This setup enables generation at the region level, where local priors act as controllable conditions for the diffusion process.

For training, the partitioning is into regions of the same shape but with randomized translation and rotation and consequently have arbitrary overlap. For sampling of the trained model, the partitioning can be on a grid which covers the area of interest (the trained-on city, the trained-on part of a city concatenated with the rest of the city, or a new city entirely) with partial border overlap.

**Canonicalization.** Each region is mapped into a canonical frame of reference using a similarity transform (Goodall, 1991), paralleling Procrustes alignment in shape analysis. Unlike group-equivariant architectures (Cohen & Welling, 2016), which encode invariances into model structure, our approach achieves invariance via input-output transformation, keeping the architecture lightweight and without additional inductive bias.

For a region $r_c$, we apply the similarity transformation $T_{r_c}(p, \alpha, s)$, parametrized by translation $p = -\mathbf{pos}(r_c)$, rotation $\alpha = -\mathbf{rot}(r_c)$, and scaling $s = 2/\mathbf{width}(r_c)$. This maps trajectories and priors into a normalized coordinate system $[-1, 1]^D$.

For example, consider a 1×1 km region in downtown Beijing: $T_{r_c}$ translates the region to the origin, rotates it to a fixed orientation, and rescales coordinates to $[-1, 1] \times [-1, 1]$. This normalization enables the model to learn local trajectory patterns (e.g., vehicles turning at intersections) that transfer across locations and cities. When sampling a region, the inverse transform is applied to each sample.

**Spatial priors.** More formally, let $x \in \mathbb{X}$ be a sample consisting of several observations $x[n] \in \mathbb{R}^D, \forall n$, where $D = 2$ denotes spatial dimensions (long, lat) We generate trajectories region by region. For each sub region $r_c$, we compute its spatial prior $H$ and express the generative model as:

$$p(x) = \int p(x|H)p(H) \, dl. \tag{1}$$

In practice, we set $p(H) = p(H = f(r_c, \mathbb{X})) = p(r_c)$ where $H$ is a discrete marginal distribution, $\sum_i \sum_j H_{i,j} = 1$, and $r_c$ is a subregion of $r$. The probability for a subregion $r_c$ is:

$$p(r_c) \propto \sum_{x \in \mathbb{X}} \sum_n \mathbb{1}(T_{r_c} x[n] \in [-1, 1]^D). \tag{2}$$

Within each region, the prior is discretized by cells $(i, j)$:

$$H_{i,j} = f(r_c, \mathbb{X})_{i,j} = \frac{\sum_{x \in \mathbb{X}} \sum_n \mathbb{1}_{r_{c_{i,j}}}(x[n])}{\sum_{x \in \mathbb{X}} \sum_n \mathbb{1}(x[n] \in \mathcal{R}_{r_c})}, \text{where} \tag{3}$$

$$\mathbb{1}_{r_{c_{i,j}}}(x[n]) = \begin{cases} 1, & \text{if } x[n] \text{ falls within cell } (i, j) \text{ of region } r_c \\ 0, & \text{otherwise} \end{cases} \tag{4}$$

In practice, we use a $64 \times 64$ grid for $3 \times 3$ km regions, balancing spatial detail with computational efficiency: finer grids increase quadratic token cost in the transformer, while coarser resolution would reduce the spatial information needed to capture detailed road structure and trajectory patterns. The final approximation for $p(x)$ is then

$$p(x) = \sum_{r_c} p(x|H = f(r_c, \mathbb{X}))p(r_c). \tag{5}$$

which is a generative mixture model over region partitions.

The spatial prior $H$ provides the context that separates *where* people move from *how* they move. By marginalizing out time, $H$ encodes only the spatial occupancy of a region. The diffusion model then learns to generate realistic temporal trajectories that, in aggregate, match this spatial pattern. Because $H$ can be estimated (even in unseen cities) for new regions (even in unseen cities), while temporal dynamics remain transferable, this factorization supports cross-region generalization. See Figure 1 for examples of spatial priors $H$ and corresponding trajectories.

To learn $p(x|H)$ we propose an architecture based on the denoising diffusion architecture (Ho et al., 2020), using a transformer encoder (Vaswani et al., 2017) for denoising.

Generating synthetic data using denoising diffusion is achieved by learning to reverse a noise-adding process. See Appendix C.1 for more details on denoising diffusion. There are several ways to parameterize the denoising process, we extend the noise prediction parameterization (Ho et al., 2020) to include the marginal distribution: $\epsilon_\theta(x_t, t, H)$. This means that at any step of the denoising process, we have full access to a discretized version of the marginal distribution of the distribution we are sampling, a noisy trajectory, as well as the expected noise level via the denoising step.

This also poses a challenge, as we need a model that can handle these different modalities. To this end, we employ a transformer encoder. This allows us to use different strategies to tokenize the

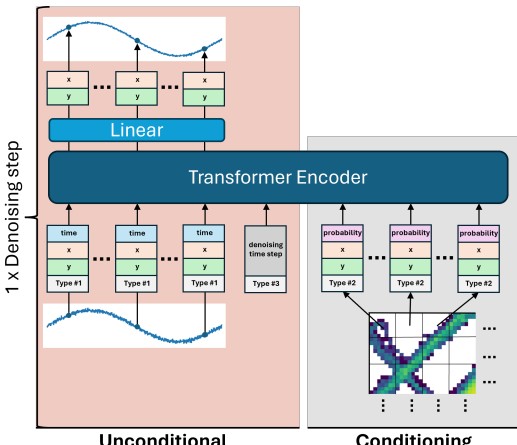

Figure 3: Architecture overview. In the unconditional part, each time point of the noisy trajectory is converted into a separate token with positional encoding used to encode its values and the time point, and a learned vector representing its type. The denoising step token, includes the denoising step encoded with positional encoding and then concatenated with a learned type vector. The input also include a marginal distribution to guide the denoising process to generate samples with particular properties, improving in-distribution performance as well as enabling generalization to previously unobserved areas. The marginal distribution is split into tokens, concatenated with a learned type vector and its position using positional encoding.

different modalities, the resulting tokens are then given as an input to the transformer. The attention mechanism allows the model to learn relevant relationship between the different token types, across different physical distances, as well as in time, to successfully denoise a trajectory.

More specifically, we employ three types of tokens: (1) *trajectory tokens*, (2) *marginal distribution tokens* and (3) *denoising noise step tokens*. At the end of each token, there is a learned token type which allows the model to distinguish between the different types. Trajectories $x_i$ are split into $L$ tokens, one for each time step, and positional encoding is used to encode position and time. For the discretized marginal distribution $H$, we follow Vision Transformers (Dosovitskiy et al., 2020) and split marginal distribution into several regions, linearly project, and finally prepend the position of the region. Finally, a single token for $t$ using positional encoding is used to denote the current step in the denoising process.

Rather than using a class-token, we use the output corresponding to the $L$ trajectory tokens to make the noise prediction. These are linearly transformed, each token down to 2 dimensions: one for x and one for y, and then concatenated across time to construct the final prediction for the amount of noise added at diffusion step $t$.

An overview of the architecture is shown in Figure 3, which illustrates how a single trajectory is tokenized, how the marginal distribution is split into subregions and which tokens are used as output. The training and sampling procedures are formalized in Algorithms 1 and 2. Algorithm 1 shows how we train the model by randomly sampling regions, computing their spatial priors, and learning to generate trajectories matching those priors. Algorithm 2 shows how we generate city-scale datasets by conditioning on spatial priors alone to enable zero-shot transfer to new regions or cities. The training and sampling procedures are shown in Algorithm 1 and 2, respectively. See Appendix C for hyperparameters, as well as additional details on tokenization.

To understand how these algorithms enable zero-shot generalization in our out-of-distribution experiments, consider the key design choices. In Algorithm 1, lines 2-3 randomly sample regions and compute spatial priors, ensuring the model encounters diverse geographic contexts during training. Line 6 canonicalizes trajectories to normalized coordinates $[0, 1]^D$, allowing the model to learn location-invariant dynamics. The denoising process (line 10) is conditioned on $H$, teaching the model to respect spatial distributional constraints. In Algorithm 2, zero-shot transfer works as follows: for target regions (unseen city areas or entirely different cities), we compute the spatial prior $H$ from

$\mathbb{X}_{\text{target}}$ trajectories (line 3), but the model $\epsilon_\theta$ never receives individual target trajectories, only their aggregate spatial distribution. The model generates trajectories in normalized space conditioned solely on $H$ (lines 7-10), then transforms them back to global coordinates of the target region (line 11). This canonicalization symmetry (normalize during training, denormalize during generation) combined with conditioning on aggregate distributions rather than trajectory instances enables the model to apply learned dynamics to any geographic region without gradient updates or fine-tuning on target data, demonstrating zero-shot generalization.

---

**Algorithm 1** Training

**Input**: Dataset $\mathbb{X}_{\text{train}}$, Region distribution $p(r_c)$, Target length
**Output:** Trained model parameters $\theta$

1: **repeat**
2: $\quad r_c \sim p(r_c)$
3: $\quad H \leftarrow f(r_c, \mathbb{X}_{\text{train}})$
4: $\quad \mathcal{X}_{r_c} \leftarrow$ Find contiguous subsequences of trajectories in $\mathbb{X}$ that lie within $r_c$
5: $\quad$ Filter $\mathcal{X}_{r_c}$ by minimum length, maximum time gaps, and speed limits
6: $\quad$ Normalize $\mathcal{X}_{r_c}$ to $[0,1]^D$ relative to $r_c$
7: $\quad$ Select random sequence $x \in \mathcal{X}_{r_c}$
8: $\quad t \sim \text{Uniform}(\{0, \ldots, T\})$
9: $\quad \epsilon \sim N(0, \mathbf{I})$
10: $\quad$ Update model parameters according to:
$\quad\quad \nabla_\theta ||\epsilon - \epsilon_\theta(\sqrt{\bar{\alpha}_t}x_0 + \sqrt{1 - \bar{\alpha}_t}\epsilon, H, t)||$
11: **until** converged

---

**Algorithm 2** Generation

**Input**: Number of total synthetic samples $N_{\text{out}}$, Target regions $\mathcal{R}_{\text{target}}$
$\quad\quad$ Dataset $\mathbb{X}_{\text{target}}$, Trained model $\epsilon_\theta$, Diffusion parameters $\{\alpha_t, \sigma_t\}_{t=1}^T$
**Output:** Synthetic trajectories $\mathcal{X}_{\text{synth}}$

1: $\mathcal{X}_{\text{synth}} \leftarrow \emptyset$
2: **for** $r_c \in R_{\text{target}}$ **do**
3: $\quad$ Compute heatmap $H = f(r_c, \mathbb{X}_{\text{target}})$
4: $\quad N_{r_c} = N_{\text{out}} \frac{\sum_x \sum_n \mathbb{1}(x[n] \in r_c)}{\sum_{\tilde{r}_c} \sum_x \sum_n \mathbb{1}(x[n] \in \tilde{r}_c)}$
5: $\quad$ **for** $n = 1, \ldots, N_{r_c}$ **do**
6: $\quad\quad x_T \sim \mathcal{N}(0, I)$
7: $\quad\quad$ **for** $t = T, \ldots, 1$ **do**
8: $\quad\quad\quad z \sim \mathcal{N}(0, I)$ if $t > 1$, else $z = 0$
9: $\quad\quad\quad x_{t-1} = \frac{1}{\sqrt{\alpha_t}}(x_t + \epsilon_\theta(x_t, H, t)) + \sigma_t z$
10: $\quad\quad$ **end for**
11: $\quad\quad$ Transform $x_0$ from $[0,1]^D$ to global coordinates of $r_c$
12: $\quad\quad \mathcal{X}_{\text{synth}} \leftarrow \mathcal{X}_{\text{synth}} \cup \{x_0\}$
13: $\quad$ **end for**
14: **end for**
15: **return** $\mathcal{X}_{\text{synth}}$

---

## 4 EVALUATION

To evaluate TDDM's trajectory generation capabilities, we address three key questions:

1. Does TDDM generate higher-quality synthetic trajectories than existing approaches?
2. Can the spatial-temporal factorization enable better coverage of complex urban distributions?
3. Does coordinate normalization actually enable generalization to unseen regions?

We start with a comparative study on the unconditional generation task and compare against leading methods representing major generative paradigms: TimeGAN (seminal time-series GAN), DiffTraj (UNet-based trajectory diffusion model), Diffusion-TS (transformer-based time-series diffusion model), TimeVAE (VAE for sequential data), and COSCI-GAN (multi-channel time-series model). These baselines provide comprehensive coverage of trajectory generation approaches. See Appendix A for detailed comparisons. There we also include theoretical comparisons with recent work without reproducible source code (TrajGen Cao & Li (2021), ControlTraj Zhu et al. (2024)) as well as with COLA Wang et al. (2024).

Then, we investigate how well TDDM can generalize to new unseen environments. First by training on a limited part of the map and, using only spatial prior, generating for the remaining map. Second by showing how a TDDM trained on one city from one dataset can generalize to another city in another dataset.

The evaluation is across three large-scale human trajectory datasets and across several evaluation measures, which together span current challenges and wanted properties of trajectory generation.

**Evaluation Measures** Several perspectives exists on synthetic data quality in the literature (Alaa et al., 2022; Wu et al., 2021; Esteban et al., 2017) covering a range of partially-overlapping aspects.

In this work, we propose to harmonize these and consequently span high fidelity, covering the support, moving beyond copying the training data, downstream task informativeness and distributional proportionality, as captured by the following five qualities: (I) *Fidelity:* (Alaa et al., 2022) The individual synthetic samples should have similar characteristics to, or be indistinguishable from, samples from the original distribution. (II) *Diversity:* (Alaa et al., 2022) It should be possible for synthetic data to be drawn from any part of the unknown distribution's support. (III) *Proportionality:* (Wu et al., 2021) The probability of a sample occurring in the synthetic distribution should be proportional to the probability of a sample occurring in the unknown distribution. (IV) *Usefulness:* (Esteban et al., 2017) The synthetic data should capture aspects of the unknown distribution that is useful for downstream tasks. (V) *Generalization:* (Alaa et al., 2022) Synthetic samples should not be mere copies of the training data.

As there is not a single measure that can encapsulate all qualities, we propose to use the following evaluation measures to paint a fuller picture of the synthetic data quality:

- **TSTR:** Train on synthetic, test on real (Esteban et al., 2017). A transformer decoder is trained on the synthetic data, with the task to predict 10 future states given a sequence of observations. The resulting model is then evaluated on the training data and the mean absolute value is reported. This evaluates the usefulness, fidelity and diversity of the synthetic data. Lower is better.
- **KL Divergence:** Evaluates support coverage and proportionality between distributions. $KL(R\|S)$ measures how well the synthetic distribution covers areas where real data exists, penalizing missed modes in the real distribution. $KL(S\|R)$ measures how well the synthetic distribution stays within regions supported by real data, penalizing unrealistic generations. Together, they provide complementary perspectives: low $KL(R\|S)$ indicates good coverage of the real distribution, while low $KL(S\|R)$ indicates high fidelity of the synthetic samples. Symmetric KL and Jensen–Shannon divergence balance these concerns, with JS providing better stability when handling regions with zero probability. Lower is better.

We also make use of the measures **Density Error**, **Trip Error**, **Length Error** and **Pattern Score**, introduced by Zhu et al. (2023). See Appendix E for details on all six measures.

**Dataset and Preprocessing**   For evaluation we use three publicly accessible real-world GPS-trajectory datasets: Geolife (Zheng et al., 2011), Porto (Moreira-Matias et al., 2013) and Cabspotting (Piorkowski et al., 2022). Geolife consists of data from 182 users performing various activities (e.g., walking, biking, driving) collected between 2007-2012, resulting in 17,621 trajectories mostly centered around Beijing, China. Porto was originally released as part of the Taxi Service Trajectory challenge and contains data on 442 taxis in Porto, Portugal, spanning more than a year. Cabspotting is also a taxi dataset, with approximately 500 taxis from the San Francisco Bay Area in California, United States, collected over 30 days. For all three datasets, we use time, longitude and latitude. The data is also resampled to one observation per second and map matching is used to reduce observation noise. All models, including baselines and TDDM, are trained and evaluated using the same preprocessed datasets. See C for more details on preprocessing.

### 4.1 LARGE-SCALE UNCONDITIONAL TRAJECTORY GENERATION

The task of unconditional trajectory generation is to learn a generative model which generate a high quality (I)-(V) synthetic dataset of trajectories. For each dataset, we train each model from scratch and then sample them to generate a synthetic dataset used for evaluation. In the case of TDDM, that means that a region and spatial prior components of the mixture are learned during training, with both relative probability between regions, $p(r_c)$, and spatial priors $H_{r_c}$ for each region partitioned on a grid covering the city in question. The sampling process is then to first draw a region, then draw a trajectory from TDDM conditioned on that region's spatial prior. Table 1 show the average performance across the three datasets.

TDDM achieves the strongest overall performance, reducing distributional divergences by a large margin compared to all baselines ($KL_{sym}$ : 0.277 vs. 1.153 for Diffusion-TS and 1.232 for DiffTraj, JS : 0.059 vs. 0.198 and 0.209 respectively). This indicates both improved coverage and better avoidance of unrealistic samples. Proportionality measures follow the same trend: TDDM attains the lowest Density and Trip errors (0.019, 0.031), whereas the next-best diffusion models remain higher

(0.029, 0.041). Fidelity is also strong: TSTR reaches 0.011, slightly improving over DiffTraj (0.013) and Diffusion-TS (0.014). On Length error, TDDM matches Diffusion-TS at state of the art (0.004 vs. 0.003), accurately capturing the distribution of distances between consecutive trajectory points (see Appendix F.2 for detailed visualizations), while all other methods trail substantially. Finally, Pattern score is highest for TDDM (0.917), confirming that it best preserves global structure. Overall, TDDM consistently surpasses existing GAN-, VAE-, and diffusion-based methods, combining high fidelity with superior distributional alignment. Results for individual cities is found in the appendix (Table 7).

Inspecting the trajectory samples visually (e.g. Figure 2) we see the advantage of TDDM over the baselines. TDDM generates the trajectories that are the most similar to the original: roads are clearly defined and there are holes in the support where no trajectories appear. Diffusion-TS, DiffTraj and TimeVAE are all capable of capturing the overall shape of the cities, but fail at generating all roads and often generates trajectories that are far from any road. TimeGAN and COSCI-GAN often fail to capture the overall chape of the cities, and TimeGAN especially struggles with mode-collapse. These observations are consistent across all three datasets, see Appendix F.1 for Geolife and Cabspotting.

## 4.2 ABLATION STUDY

To probe the role of spatial priors and region size, we conduct ablations (Table 2). Removing spatial priors leaves TSTR unchanged but degrades KL-based scores by up to 5 times, showing that temporal dynamics alone provide useful signals but fail to ensure coverage and proportionality. Reducing partition size to $1 \times 1$ km slightly increases Pattern (0.930 vs. 0.917) but worsens Length error (0.150 vs. 0.004), revealing a tradeoff between local coherence and global realism. The full $3 \times 3$ km prior-based model provides the best overall balance. Results for individual cities is found in the appendix (Table 8).

Map matching is used as part of the preprocessing to align the raw GPS trajectories to maps for the respective cities, before GPS noise is added back. The same preprocessed data is used throughout the experiments, both for training models and as the target distribution for the evaluation. To verify the effect of map-matching, the top-three models (TDDM, Diffusion-TS, DiffTraj) are also trained without map matching. All models show a significant drop in performance when not using map-matched data, especially for Cabspotting due to its lower frequency of position updates. The results, shown in Table 9 in the Appendix is consistent with the results where map-matching is used. This demonstrates that TDDM's improvements in performance compared to the baselines stem from the deaggregation framework.

## 4.3 OUT-OF-DISTRIBUTION GENERALIZATION

Finally, we evaluate zero-shot generalization (Table 3) for *intra-city* and *city-to-city* transfer. In both settings, models are trained only on source regions and generate for target regions using solely the spatial prior $H$, with no gradient updates on target trajectories. Results for individual cities, including

Table 1: Evaluation of different models' performance across several datasets and measures. Models are trained, sampled and evaluated once per dataset. The results are then averaged across datasets.

| Measure | TimeGAN | TimeVAE | COSCI-GAN | Diffusion-TS | DiffTraj | TDDM |
|---|---|---|---|---|---|---|
| TSTR ($\downarrow$) | $0.037 \pm 0.027$ | $0.018 \pm 0.010$ | $0.023 \pm 0.007$ | $0.014 \pm 0.009$ | $\underline{0.013 \pm 0.005}$ | $\mathbf{0.011 \pm 0.006}$ |
| KL($S \parallel R$) ($\downarrow$) | 3.702 | 2.363 | 3.046 | $\underline{1.395}$ | 1.594 | **0.301** |
| KL($R \parallel S$) ($\downarrow$) | 2.586 | 1.268 | 1.740 | 0.911 | $\underline{0.869}$ | **0.253** |
| KL$_{\text{sym}}$ ($\downarrow$) | 3.144 | 1.816 | 2.393 | $\underline{1.153}$ | 1.232 | **0.277** |
| JS ($\downarrow$) | 0.397 | 0.287 | 0.363 | $\underline{0.198}$ | 0.209 | **0.059** |
| KL$_{\text{speed}}$ ($\downarrow$) | 0.465 | 0.225 | 6.463 | $\underline{0.035}$ | 0.126 | **0.013** |
| Density ($\downarrow$) | 0.258 | 0.043 | 0.134 | $\underline{0.029}$ | 0.033 | **0.019** |
| Trip ($\downarrow$) | 0.323 | 0.056 | 0.158 | $\underline{0.041}$ | 0.042 | **0.031** |
| Length ($\downarrow$) | 0.097 | 0.042 | 0.789 | **0.003** | 0.065 | $\underline{0.004}$ |
| Pattern ($\uparrow$) | 0.677 | 0.840 | 0.770 | $\underline{0.907}$ | 0.893 | **0.917** |

Table 2: Ablation study. Models are trained, sampled and evaluated once per dataset. The results are then averaged across datasets. Note that TDDM has a $3 \times 3$ km region size.

| Measure | TDDM | 1x1 km | w/o spatial prior | w/o spatial prior + rejection |
|---|---|---|---|---|
| TSTR ($\downarrow$) | **0.011 ± 0.006** | 0.024 ± 0.012 | **0.011 ± 0.006** | 0.014 ± 0.010 |
| KL($S \parallel R$) ($\downarrow$) | **0.301** | 0.339 | 1.569 | 1.925 |
| KL($R \parallel S$) ($\downarrow$) | **0.253** | 0.318 | 1.098 | 1.252 |
| KL$_{sym}$ ($\downarrow$) | **0.277** | 0.328 | 1.334 | 1.588 |
| JS ($\downarrow$) | **0.059** | 0.071 | 0.228 | 0.266 |
| KL$_{speed}$ ($\downarrow$) | **0.013** | 0.583 | 0.323 | 0.422 |
| Density ($\downarrow$) | **0.019** | 0.022 | 0.067 | 0.063 |
| Trip ($\downarrow$) | **0.031** | 0.044 | 0.074 | 0.081 |
| Length ($\downarrow$) | **0.004** | 0.150 | 0.078 | 0.075 |
| Pattern ($\uparrow$) | 0.917 | **0.930** | 0.833 | 0.860 |

the full city-to-city transfer table is found in the appendix (Table 12). Visualization of the marginal distribution of the synthetic datasets are shown in Appendix (Figure 15).

Zero-shot intra-city transfer, where the model is trained from scratch on only 25% of a map (specifically, a geographically contiguous quadrant as shown in Figures 12–14) and applied to the rest, shows that aggregated TSTR remains comparable to full coverage (0.010 vs. 0.010), while KL$_{sym}$ and JS divergences increase (0.545 vs. 0.278; 0.106 vs. 0.059). Pattern also remains high (0.927 vs. 0.940), indicating that spatial priors act as a strong regularizer. Although these aggregated results suggest comparable performance, per-city analysis in the Appendix reveals more substantial variations (Table 12).

In zero-shot city-to-city transfer, performance varies with the source dataset but remains competitive: Pattern stays above 0.915 across all cases, proportionality measures are consistently lower than GAN/VAEs and close to diffusion baselines, and TSTR often matches in-distribution performance when trained on Porto. The main weakness is Length error, which increases to 0.06–0.11 across cities, suggesting that fine-grained distance modeling is less transferable. Despite this, TDDM demonstrates robust fidelity and distributional generalization across cities, highlighting the effectiveness of separating spatial priors from temporal dynamics.

Interestingly, cross-city transfer from Porto often yields stronger results than training on limited portions of the target city. On average, models trained on Porto generalize with lower KL and JS divergences (0.335 and 0.071) than those trained on only 25% of the target city (0.545 and 0.106), and they also maintain slightly better proportionality and Pattern scores (0.930 vs. 0.927). The only exception is Length error, where access to even a small fraction of local data provides an advantage (0.026 vs. 0.060). This reflects city-specific differences in trajectory length distributions, which cannot be inferred from spatial priors alone. For instance, Porto exhibits a heavier-tailed distribution than Cabspotting (Appendix F.2, Figure 19). This suggests that Porto captures temporal dynamics and spatial statistics that are broadly representative across cities, making it an unexpectedly strong *universal source* dataset for this setting. More generally, these results highlight a tradeoff: if path-length accuracy is paramount, local data (even in small amounts) remains valuable, but for distributional coverage and spatial structure, carefully chosen training cities may outperform partial local coverage.

Table 3: Generalization performance across different training scenarios. Left: Intra-city generalization comparing training on the partial (25%) versus full (100%) spatial domain. Right: City-to-city generalization where models are trained on one dataset and evaluated on others. Results are aggregated across datasets.

| Measure | Intra-city (Training data) | | | City-to-city (Trained on) | |
| | 25% | 100% | Geolife | Porto | Cabspotting |
| --- | --- | --- | --- | --- | --- |
| TSTR ($\downarrow$) | **0.010 ± 0.006** | **0.010 ± 0.007** | 0.016 ± 0.008 | **0.010 ± 0.005** | 0.011 ± 0.006 |
| KL($S \parallel R$) ($\downarrow$) | 0.615 | **0.305** | 0.903 | **0.357** | 0.610 |
| KL($R \parallel S$) ($\downarrow$) | 0.474 | **0.251** | 0.688 | **0.313** | 0.449 |
| KL$_{\mathrm{sym}}$ ($\downarrow$) | 0.545 | **0.278** | 0.795 | **0.335** | 0.530 |
| JS ($\downarrow$) | 0.106 | **0.059** | 0.149 | **0.071** | 0.102 |
| KL$_{\mathrm{speed}}$ ($\downarrow$) | 0.101 | **0.012** | 0.322 | **0.238** | 0.393 |
| Density ($\downarrow$) | 0.021 | **0.015** | **0.018** | **0.018** | 0.022 |
| Trip ($\downarrow$) | 0.036 | **0.027** | **0.031** | 0.036 | 0.042 |
| Length ($\downarrow$) | 0.026 | **0.003** | 0.082 | **0.060** | 0.109 |
| Pattern ($\uparrow$) | 0.927 | **0.940** | 0.925 | **0.930** | 0.915 |

# 5 CONCLUSION

We have presented the Temporal Deaggregation Diffusion Model (TDDM), a hierarchical generative framework that separates spatial priors from temporal dynamics for large-scale trajectory generation. Across three major urban datasets, TDDM consistently improves distributional alignment, achieving up to 4 times lower KL divergences than the best diffusion baselines, while also setting new state of the art on Density, Trip, and Pattern measures. Importantly, TDDM matches leading models on Length error and outperforms them on fidelity as measured by TSTR. This is also confirmed visually.

Beyond unconditional generation, ablation studies highlight the critical role of spatial priors for coverage and proportionality, while generalization experiments demonstrate that TDDM can synthesize realistic trajectories in unseen parts of cities and across entirely new cities. In particular, we find that training on Porto generalizes better on average to other cities than training on partial local data, suggesting that certain cities may act as representative source datasets. That is, datasets that generalize broadly across urban contexts and provide stronger transferability than limited amounts of local data.

Taken together, these results show that factorizing *where* and *how* people move not only advances generation quality but also unlocks strong out-of-distribution generalization. TDDM thus provides both a methodological advance and a practical step toward scalable and generalizable mobility modeling.

**Future Work.** Several promising directions remain for extending TDDM. First, augmenting the spatial prior $H$ with additional marginal information (such as trajectory length distributions, time-of-day priors, or directional priors for traffic handedness) could improve temporal fidelity and enable accurate generalization to left-hand vs. right-hand traffic patterns. Second, incorporating road hierarchy information as priors would better capture local speed structure and road network topology. Finally, extending $H$ with city-specific temporal marginals would likely improve cross-city temporal fidelity while preserving the zero-shot spatial generalization capabilities demonstrated in this work.

ETHICS STATEMENT

This work focuses on improving the quality and generalization capabilities of synthetic human mobility trajectory generation using publicly available GPS datasets (Geolife, Porto, Cabspotting) with appropriate licenses as detailed in Appendix B.1. We acknowledge that synthetic trajectory generation technology has dual-use potential. While our primary motivation is to advance the state-of-the-art in generative modeling to enable beneficial applications such as urban planning, transportation research, and mobility simulation, we recognize that high-quality synthetic data generation techniques could potentially be misused for surveillance or other harmful purposes. We encourage responsible use of our methods and emphasize that practitioners should consider appropriate safeguards when deploying synthetic trajectory generation in real-world applications. Additionally, we note that privacy considerations must be carefully evaluated when applying our methods, as even publicly available trajectory datasets can potentially be used for re-identification when combined with other data sources or when analyzed with sufficient temporal and spatial resolution. We emphasize that our methods do not guarantee privacy-preserving synthetic data generation, and we strongly encourage practitioners to combine our approach with established privacy-preserving techniques such as differential privacy when working with real-world trajectory data. Our research contributes to the broader goal of developing high-fidelity generative models that can support legitimate research and planning activities while reducing the need for access to sensitive real trajectory data. We have designed our evaluation framework to focus on distributional properties and generation quality without enabling inference about specific individuals in the original datasets.

REPRODUCIBILITY

To ensure full reproducibility of our results, we provide comprehensive implementation details and experimental resources. We release complete, runnable source code for all experiments [1], including implementations of TDDM and all baseline methods, along with evaluation code for all measures used. The core TDDM algorithm is detailed in Algorithms 1 and 2, with complete architectural specifications in Section 3 and extended implementation details in Appendix C. All hyperparameters used in our experiments are provided in Appendix C.4, including the search space and final values obtained through systematic optimization. Dataset preprocessing steps for all three datasets (Geolife, Porto, Cabspotting) are comprehensively described in Appendix B, including data licenses, filtering criteria, and map-matching procedures. Our evaluation methodology is detailed in Section 4 with extended measure descriptions in Appendix E. All experimental configurations, including baseline implementations and evaluation protocols, are provided to enable direct replication of our results across all datasets and experimental conditions.

LLM USAGE

This paper was developed with assistance from large language models as writing and research tools, including Claude (Anthropic), ChatGPT (OpenAI), and Cursor AI for code assistance. The LLMs contributed to:

- **Writing refinement**: Improving clarity, flow, and exposition throughout the paper
- **Technical positioning**: Helping articulate the methodological approach and frame the contribution within existing literature
- **Experimental analysis**: Assisting with evaluation design and results presentation
- **Literature review**: Supporting the discovery and organization of related work
- **Code development**: Assisting with implementation and debugging of code

All research concepts, experimental design, implementation, and scientific conclusions are the original work of the authors. All LLM-generated content has been verified and validated by the authors. The authors take full responsibility for all content and claims presented in this paper.

---

[1]Available at https://anonymous.4open.science/r/tddm/.

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

APPENDIX: SUPPLEMENTARY MATERIAL FOR LEARNING TO DEAGGREGATE

The appendix contains supplementary material to the paper *Learning to Deaggregate: Unconditional Generation of Trajectories at Scale*. We provide detailed information on related work, model architecture, implementation details, extended experimental results, and additional visualizations.

# Contents

# A EXTENDED RELATED WORK

In Section 1 and 4 of the main paper, we described the state-of-the-art models used for comparison. Here, we provide extended details on how these models differ from our proposed TDDM approach, particularly regarding their conditional generation capabilities and architectural choices.

Previous work on unconditional generation of time-series data has focused on variations of the generative adversarial networks (GAN) architecture (Esteban et al., 2017; Yoon et al., 2019a; Jeon et al., 2022) and, more recently diffusion models (Zhu et al., 2023; Yuan & Qiao, 2024).

There has also been an interest in using time-series generation for imputation and forecasting (Alcaraz & Strodthoff, 2023; Tashiro et al., 2021; Shen & Kwok, 2023; Dai et al., 2023; Feng et al., 2024). Transformer-based time-series foundation models have been proposed as a general purpose forecasting tool (Ansari et al., 2024), but has not been evaluated on the unconditional generation task.

TimeGAN (Yoon et al., 2019a) consists of a generative adversarial network (GAN) operating inside the latent space of an autoencoder. To further improve performance, they add an additional network with the task of predicting one time step ahead. The encoder, decoder, supervisor, generator and discriminator are all implemented using autoregressive models and in practice they use gated recurrent units (GRUs).

The TimeVAE (Desai et al., 2021) architecture is a variant of the popular variational autoencoder architecture. The autoencoder is trained with an additional loss component to have the latent space conform to a known statistical distribution, in this instance a multivariate normal distribution. The

autoencoder is trained to both minimize the reconstruction loss, as well as minimizing the divergence between the embedded data and the prior set for the latent space.

COSCI-GAN (Seyfi et al., 2022) proposes to use a separate generative adversarial network for each channel of the data. The individual GANs all share single source of noise as input to the generator and, additionally they have a central discriminator that is given the stacked output from the all the generators as input.

Diffusion-TS (Yuan & Qiao, 2024) adapts the denoising diffusion architecture (Ho et al., 2020) to generate time-series data by implementing the denoising step with a multilayer neural network each consisting of a transformer block, a fully connected neural network as well as time-series specific layers with the aim of improved performance and interpretability. The architecture also has support for conditional generation to enable using the models for imputation and forecasting. As Diffusion-TS moves away from the U-Net approach used in trajectory-specific models to a partial Transformer architecture, we extend this further to a full Transformer-based architecture, avoiding additional time-series specific induction biases while gaining increased performance for synthetic data generation.

Recent work has extended these generative architectures to the specific domain of GPS trajectory generation, with varying approaches to conditional control.

DiffTraj (Zhu et al., 2023) adapts the denoising diffusion architecture (Ho et al., 2020) to generate GPS-trajectories. The U-Net architecture used in (Ho et al., 2020) for denoising, is also adapted to work with trajectories. As part of the generation process, the model is conditioned on individual trajectory-specific statistics such as velocity, distance, and departure time.

ControlTraj (Zhu et al., 2024) is a diffusion-based framework for generating GPS trajectories conditioned on road network topology from OpenStreetMap and trip-specific attributes (departure time, distance, speed). Similar to DiffTraj, ControlTraj uses trajectory-level conditioning to control individual samples. In contrast, TDDM conditions on spatial priors (aggregate marginal occupancy distributions computed from multiple trajectories).

COLA (Wang et al., 2024) is a model for generating discrete trajectories, i.e., where space is discretized into location IDs, and where samples are temporally sparser (hourly timesteps vs. 1-second for TDDM). Both COLA and TDDM address city-to-city transfer, but using different paradigms. COLA uses transfer learning with gradient-based adaptation, while TDDM achieves zero-shot transfer via spatial prior conditioning. The zero-shot approach only requires marginal occupancy information, while the transfer learning approach requires access to full trajectories from the target domain.

TrajGen (Cao & Li, 2021) is an unconditional generative model for trajectories. It uses an unconditional generative image generative model to generate images of trajectories. The images show a top-down view of single trajectories. These are then decoded into a sequence of points, which are then ordered and then timestamped. While both TrajGen and TDDM separate spatial and temporal information, they differ fundamentally in representation and conditioning: TrajGen's unconditional approach means that once trained, the model has limited control of what trajectory patterns are generated, whereas TDDM explicitly conditions on spatial priors H, enabling controllable generation and zero-shot cross-city transfer.

While DiffTraj, ControlTraj, and Diffusion-TS all support conditional generation, our approach differs fundamentally in the type and purpose of conditioning information. DiffTraj and ControlTraj condition on sample-specific statistics (velocity, distance, road networks), creating a near one-to-one mapping between conditions and individual trajectories. Diffusion-TS uses conditioning for enabling imputation and forecasting rather than improving generation quality. In contrast, TDDM conditions on spatial priors (aggregate marginal occupancy distributions computed from multiple trajectories), which cannot be directly tied to any individual sample. This distinction enables TDDM to generalize to entirely new environments through zero-shot transfer (requiring only marginal occupancy information), while sample-specific conditioning limits generation to the trained spatial region and increases memorization risk. TrajGen's unconditional approach offers no control over generated patterns, while COLA's transfer learning paradigm requires access to full trajectories from the target domain for gradient-based adaptation.

## B    DATASETS AND PREPROCESSING

In this section, we describe the datasets used in our experiments and the preprocessing steps applied to prepare the data for training and evaluation.

The datasets described in Section 4 of the main paper provide geographical diversity across different continents while ensuring sufficient data density for meaningful evaluation. Our preprocessing pipeline, described below, is applied uniformly to all methods to ensure fair comparison.

The trajectory segmentation process helps all methods by creating cleaner, more consistent training data with fewer anomalies. It is important to note that while our model operates on such local regions during both training and inference, we do not generate trajectories that extend beyond the size of regions used during training. Instead, the hierarchical approach allows us to generate coherent trajectories for multiple adjacent regions that together compose large-scale environments.

### B.1    DATASET DESCRIPTIONS AND LICENSES

All three datasets used in this work are publicly available and contain GPS trajectory data from different geographical regions:

- **Geolife** (Zheng et al., 2011): Data from 182 users collected between 2007-2012, resulting in 17,621 trajectories mostly centered around Beijing, China. Released under the Microsoft Research License Agreement, which allows academic research but prohibits redistribution.
- **Porto** (Moreira-Matias et al., 2013): Released as part of the Taxi Service Trajectory challenge, containing data on 442 taxis in Porto, Portugal, spanning more than a year. Released under the CC BY 4.0 license.
- **Cabspotting** (Piorkowski et al., 2022): Taxi dataset with approximately 500 taxis from the San Francisco Bay Area in California, United States, collected over 30 days. Released under the CC BY 4.0 license. An account is required to download this data.

For all three datasets, we use time, longitude and latitude as the primary features.

### B.2    PREPROCESSING

Raw GPS data contains multiple problems:

- GPS drift
- GPS being turned on even when the vehicle is not in use
- large spikes in velocity

We split trajectories into sub-trajectories if any observation leaves the geographic bounds, exceeds the velocity limit or if too long time has passed since the previous observation. In the raw data, a trajectory can span several days with several hours between observations. Introducing a time-limit allows us to break these trajectories into individual and more time-constrained journeys.

We up-sample the data to one observation per second. Map matching using Valhalla (Valhalla Contributors, 2025) is then used to map the GPS trajectories to the road network, the distance between the matched and interpolated trajectories are used to estimate a noise distribution. The final training data consists of the map matched trajectories with trajectory-noise added from the noise distribution, i.e. we sample the noise distribution once per trajectory.

### B.3    HARDWARE

All experiments were conducted on a system with the following specifications:

- **CPU**: AMD Ryzen 9 5900X
- **RAM**: 128 GB
- **GPU**: NVIDIA GeForce RTX 3090 Ti (24 GB GDDR6X)

# C  ARCHITECTURE DETAILS

## C.1  DENOISING DIFFUSION PROBABILISTIC MODELS

Denoising diffusion probabilistic includes two processes: a known noise-adding process and an approximated denoising process. The noise-adding process starts at a random sample $x \in \mathbb{X}$, denoted $x_0$ where 0 is the noise step, and adds noise at each steps according to a noise schedule $\beta = \{\beta_1, \ldots, \beta_T\}$. By using a Gaussian noise process (Ho et al., 2020), the forward process has a closed expression for time point $t$: $q(x_t | x_0) = \mathcal{N}(x_t; \sqrt{\bar{a}_t} x_0, (1 - \bar{\alpha}_t) I)$, where $\alpha_t := 1 - \beta_t$, $\bar{\alpha}_t := \prod_{s=1}^{t} \alpha_s$ . The reverse process is starts at $p(x_T) = \mathcal{N}(x_T; 0, I)$ and the transition function for $1 < t \leq T$ is learned:

$$p_\theta(x_{t-1} | x_t) = \mathcal{N}(x_{t-1}; \mu_\theta(x_t, t), \Sigma_\theta(x_t, t)) \tag{6}$$

There are different ways to parameterize $\mu_\theta(x_t, t)$ and $\Sigma_\theta(x_t, t)$ and we use the noise prediction parameterization (Ho et al., 2020):

$$\mu_\theta(x_t, t) = \frac{1}{\sqrt{\alpha_t}} \left( x_t - \frac{\beta_t}{\sqrt{1 - \bar{\alpha}_t}} \epsilon_\theta(x_t, t) \right), \qquad \Sigma_\theta(x_t, t) = \sigma_t^2 I \tag{7}$$

where $\epsilon_\theta(x_t, t)$ is a learned function and is trained to predict the added noise at step $t$ of the noise adding process. In this work, we extend $\epsilon_\theta$ with discretized marginal distribution $H$ as described in the main paper in Section 3.

## C.2  OPTIMIZATION DETAILS

We use the following positional encoding to encode a one-dimensional signal (Vaswani et al., 2017):

$$\text{PE}_{(pos, 2i)} = \sin \left( -e^i \frac{\log(10000)}{\frac{d}{2} - 1} \right) \tag{8}$$

$$\text{PE}_{(pos, 2i+1)} = \cos \left( -e^i \frac{\log(10000)}{\frac{d}{2} - 1} \right), \text{where} \tag{9}$$

pos is the signal, e.g. a position $x$, position $y$ or time point $t$, and $i$ is the target dimension. For position, we encode a single position observation in $\mathbb{R}$ into $\mathbb{R}^D$, so the function would be called once for $i \in 1 \ldots D$. We use the cosine noise schedule as proposed by Nichol & Dhariwal (2021):

$$\bar{\alpha}_t = \frac{f(t)}{f(0)}, f(t) = \cos \left( \frac{t/T + s}{1 + s} \cdot \frac{\pi}{2} \right)^2 \tag{10}$$

with $s = 0.008$. Furthermore, we use the simplified loss function proposed by Ho et al. (2020):

$$\mathcal{L}_{\text{simple}}(\theta) := \mathbb{E}_{t, x_0, \epsilon} \left[ \left|\left| \epsilon - \epsilon_\theta(\sqrt{\bar{\alpha}_t}, x_0 + \sqrt{1 - \bar{\alpha}_t} \epsilon, t) \right|\right|^2 \right] \tag{11}$$

We use the Adam (Kingma & Ba, 2014) optimizer and implement our model in PyTorch (Ansel et al., 2024).

## C.3  ENCODER INPUT

We set $N$ to be a hyperparameter, and $D$ is the number of dimensions of a single observation in a trajectory, in our case $D = 2$. The input to the transformer encoder is:

- L input tokens, each token corresponding to a time point in the noisy sequence. **Note:** The token size depends on the number of features of the dataset. It is a concatenation of:
    - $x \in \mathbb{R}^{DN}$, each corresponding to a dimension observed at each time point encoded using positional encoding
    - $x \in \mathbb{R}^N$, the time point encoded using the positional encoding introduced in (Vaswani et al., 2017)

– $x \in \mathbb{R}^N$, a learned vector encoding denoting that this is a token that corresponds to a noisy sequence

- **Conditional information:** 64 tokens, each corresponding to a patch of the heatmap and being a concatenation of:

  – $x \in \mathbb{R}^N$, corresponding to the x position of the patch. Encoded using positional encoding (Vaswani et al., 2017).

  – $x \in \mathbb{R}^N$, corresponding to the y position of the patch. Encoded using positional encoding (Vaswani et al., 2017).

  – $x \in \mathbb{R}^N$, corresponding to the intensity of the heatmap. Encoded using a linear projection (Dosovitskiy et al., 2020).

  – $x \in \mathbb{R}^N$, a learned vector encoding denoting that this is a token that corresponds to the conditional information

- A token encoding the current denoising step:

  – $x \in \mathbb{R}^{N(D+1)}$, the denoising step encoded using positional encoding (Vaswani et al., 2017)

  – $x \in \mathbb{R}^N$, a learned vector encoding denoting that this is a token that corresponds to the denoising step

## C.4 HYPERPARAMETERS

We employ Optuna (Akiba et al., 2019) with Tree-structured Parzen Estimator (TPE) sampling to optimize our model's hyperparameters. The optimization process consists of 50 trials, where the model is trained from scratch trains for 100 epochs. For each trial, we sample hyperparameters from the defined search space and evaluate the model's performance by calculating the Jensen-Shannon divergence between the marginal distributions of the synthetic data and the marginal distributions of the training data. We only use the Geolife dataset for hyperparameter tuning. The hyperparameters included in the optimization are shown in Table 4 and the final values are shown in Table 5.

Table 4: Hyperparameter Search Space

| Parameter | Search Space | Description |
|---|---|---|
| Hidden Dimension | $\{4, 8, 16, 32, 64, 128\}$ | Size of transformer hidden layers |
| Number of Layers | $\{1, 2, 4, 8\}$ | Depth of transformer architecture |
| Number of Attention Heads | $\{1, 2, 4, 8\}$ | Multi-head attention mechanism size |
| Diffusion Timesteps | $\{100, 500, 1000\}$ | Number of diffusion steps |
| Learning Rate | $[10^{-6}, 10^{-2}]$ | Log-uniform sampling |

The Jensen-Shannon divergence was chosen as the optimization metric because it directly aligns with our goal of generating synthetic trajectories that match the distributional properties of real data, as discussed in Section 4 where we evaluate models on both sample-level fidelity and distribution-level similarity.

Table 5: Final Hyperparameters

| Parameter | Value |
|---|---|
| Hidden Dimension | 128 |
| Number of Layers | 8 |
| Number of Attention Heads | 2 |
| Diffusion Timesteps | 500 |
| Learning Rate | 0.00017483 |

## D  COMPUTATIONAL COMPLEXITY AND RUNTIME ANALYSIS

In this section, we provide a detailed analysis of TDDM's computational complexity and empirical runtime characteristics. This addresses concerns about the scalability and practical deployment of our method for large-scale trajectory generation.

### D.1  THEORETICAL COMPLEXITY

The computational complexity of TDDM is determined by the transformer encoder architecture used for denoising at each diffusion step. For a single denoising step, the per-trajectory cost is:

$$O((L + R)d^2 + (L + R)^2 d) \tag{12}$$

where:

- $L$ is the trajectory sequence length (number of time steps)
- $R$ is the number of spatial prior patches (64 in our implementation)
- $d$ is the hidden dimension of the transformer (128 in our final model)

The first term $(L + R)d^2$ corresponds to the cost of the feed-forward layers in the transformer, while the second term $(L + R)^2 d$ corresponds to the cost of the self-attention mechanism that processes both trajectory tokens and spatial prior tokens jointly.

**Scalability to large cities.**  Importantly, because we operate on fixed-size $3 \times 3$ km regions with a constant number of spatial prior patches ($R = 64$), the per-trajectory computational cost remains *constant* across cities of different sizes. City-scale generation scales *linearly* with geographic coverage (city area), as we independently process each region and stitch the results together according to the mixture model formulation in Section 3. This is in contrast to methods that must process entire city-scale contexts, which would scale quadratically with area in attention-based models.

### D.2  EMPIRICAL RUNTIME AND MEMORY REQUIREMENTS

We measure TDDM's inference performance on an NVIDIA GeForce RTX 3090 Ti (24 GB GDDR6X) for different batch sizes. Table 6 shows the time per trajectory and peak GPU memory consumption.

Table 6: TDDM inference performance across different batch sizes on RTX 3090 Ti. Time per trajectory decreases with larger batch sizes due to improved GPU utilization, while memory consumption grows sub-linearly.

| Batch Size | Time per Trajectory (ms) | Peak GPU Memory (MB) |
|:---:|:---:|:---:|
| 1 | 1162 | 313 |
| 8 | 427 | 340 |
| 64 | 349 | 558 |
| 256 | 340 | 1307 |

Key observations from these measurements:

- **Batching efficiency**: Increasing batch size from 1 to 256 reduces per-trajectory time by approximately 3.4× (from 1162 ms to 340 ms), demonstrating good GPU utilization at higher batch sizes.
- **Memory efficiency**: Peak GPU memory grows sub-linearly with batch size. A 256× increase in batch size only requires approximately 4.2× increase in memory (from 313 MB to 1307 MB), indicating efficient memory usage that can accommodate large-scale generation on modern GPUs.
- **Practical throughput**: At the batch size of 256, TDDM can generate approximately 2.9 trajectories per second on a single RTX 3090 Ti. For a typical city-scale dataset with 50,000 trajectories, complete generation takes approximately 4.7 hours.

- **Comparison with baselines**: While direct runtime comparisons depend on implementation details, hardware, and sampling procedures, TDDM's per-trajectory inference time of 340ms is competitive with other diffusion-based methods. The ability to batch efficiently and the constant per-trajectory cost (independent of city size) make TDDM practical for large-scale applications.

# E  EXTENDED EVALUATION MEASURES DESCRIPTION

In Section 4, we introduced evaluation measures spanning multiple quality dimensions for synthetic trajectory data. Here we provide a more detailed description of these measures, their theoretical foundations, and how they complement each other in our benchmark.

## E.1  DIFFTRAJ MEASURES

The measures from DiffTraj (Zhu et al., 2023) are[2]:

- **Density Error**: A pair of heatmaps of the training and synthetic data are calculated by dividing the city into $16 \times 16$ blocks. The number of observations in each block are counted and normalized. The Jensen-Shannon divergence is calculated between the training data heatmap and synthetic data heatmap.

- **Trip Error**: Two heatmap pairs, each $16 \times 16$ blocks, are calculated. The first pair is of the start positions for all trajectories, one heatmap for the training data and another for the synthetic data. The second pair is calculated from the last position for all trajectories. The Jensen-Shannon divergence is calculated once for each pair and then the average is reported.

- **Length Error**: The distance between consecutive observations are calculated, once for the training data and once for the synthetic data. Histograms are calculated for each, with the number of bins set to 16. Finally, the Jensen-Shannon divergence is calculated between the histogram of training data and the histogram of the synthetic data.

- **Pattern Score**: Using the heatmaps from Density Error, the top N areas (highest count) from the training and synthetic data are collected. The F-score is then calculated and reported.

## E.2  KL-BASED DISTRIBUTION MEASURES

Our evaluation framework incorporates several Kullback-Leibler (KL) divergence-based measures that address different quality dimensions of synthetic trajectory data:

- **KL(R‖S)**: Kullback-Leibler divergence of the real (R) distribution from the synthetic (S) distribution measures how well the synthetic distribution covers areas where real data exists:

$$\mathrm{KL}(R \parallel S) = \sum_{i,j} R_{i,j} \log \frac{R_{i,j}}{S_{i,j}} \tag{13}$$

  This directly corresponds to our *diversity* quality dimension (II), as it heavily penalizes when synthetic data misses modes present in the real distribution.

- **KL(S‖R)**: Kullback-Leibler divergence of the synthetic (S) distribution from the real (R) distribution measures how well the synthetic distribution stays within regions supported by real data:

$$\mathrm{KL}(S \parallel R) = \sum_{i,j} S_{i,j} \log \frac{S_{i,j}}{R_{i,j}} \tag{14}$$

  This reflects our *fidelity* quality dimension (I), penalizing synthetic data that generates unrealistic trajectories in regions with little or no real data.

---

[2]To the best of the authors ability to interpret the paper since the evaluation code is not available for DiffTraj.

- **KL$_{\text{sym}}$**: The symmetric KL divergence balances both directional measures:

$$\text{KL}_{\text{sym}} = \frac{1}{2}(\text{KL}(S \parallel R) + \text{KL}(R \parallel S)) \tag{15}$$

- **JS**: Jensen-Shannon divergence provides a bounded symmetric measure:

$$\text{JS}(S, R) = \frac{1}{2}\text{KL}(S \parallel M) + \frac{1}{2}\text{KL}(R \parallel M) \tag{16}$$

where $M = \frac{1}{2}(S + R)$. This avoids numerical instabilities with zero probabilities and is bounded between 0 and 1.

Together, these measures address the *proportionality* quality dimension (III) - low values in both KL directions indicate the synthetic distribution not only covers the same support as the real distribution but also assigns similar probability mass across that support. The symmetric measures (KL$_{\text{sym}}$ and JS) provide balanced assessments that account for both support coverage and proportionality, with JS offering better numerical stability when regions with zero probability are present in either distribution.

### E.3 THEORETICAL JUSTIFICATION AND IMPLEMENTATION

Our implementation of KL-based measures uses a discretizations with a $256 \times 256$ grid over the entire city. This is $\times 256$ higher resolution than what is used in DiffTraj, to enabling more precise evaluation of fine-grained spatial patterns.

KL divergence measures are theoretically well-suited for trajectory evaluation because:

- They directly quantify the information loss when approximating one distribution with another
- They are sensitive to both the support coverage and the proportionality of distributions
- The two directional variants (KL(S∥R) and KL(R∥S)) provide complementary insights into different failure modes

### E.4 RELATION TO SAMPLE QUALITY

The combined set of measures provides a comprehensive evaluation framework addressing all five quality dimensions from Section 4:

- **Fidelity**: Length Error, Trip Error, and KL(R∥S) all measure aspects of fidelity by ensuring trajectories have realistic properties.
- **Diversity**: KL(S∥R) and Pattern Score evaluate how well synthetic trajectories cover the support of real data.
- **Proportionality**: All KL-based measures, but especially JS, capture proportionality by measuring distributional similarity.
- **Usefulness**: TSTR directly measures usefulness for a downstream prediction task.
- **Generalization**: See Section E.6.

### E.5 LIMITATIONS ANALYSIS

The DiffTraj measures provide a more coarse-grained evaluation using $16 \times 16$ spatial discretization, while our KL-based measures use $256 \times 256$ grids for finer-grained assessment. The $\times 256$ higher resolution allows more precise evaluation of the model's ability to capture detailed spatial distributions, especially important in urban environments where road networks create complex movement patterns.

By combining these measures with TSTR, our benchmark provides a comprehensive evaluation that addresses both distribution-level properties and practical utility of the generated trajectories, spanning all important dimensions of synthetic data quality discussed in Section 4.

### E.6 Evaluating Generalization

Our evaluation framework extends beyond traditional in-distribution testing to assess different types of generalization capabilities:

- *In-distribution generalization*: Unlike predictive models where test sets evaluate prediction accuracy, unconditional generative models require evaluating whether they capture the full distribution rather than memorizing training examples. For trajectory generation, we directly compare distribution-level properties between synthetic and real data instead of using held-out test sets. Using a held-out test would require massive data amounts and/or an inverted data split, e.g. 20% train, 80% test to sufficiently assess support coverage and proportionality. This is currently infeasible in practice. Also, current models still struggle to even replicate (memorize) the training data in unconditional generation, making this less of a relevant problem in the field for now.

- *Out-of-distribution generalization*: A distinctive feature of our benchmark is its assessment of spatial transfer capability, evaluating a model's ability to generate plausible trajectories for geographic regions not represented in training data:
    - *Intra-city generalization* (Section 4.3): Tests whether models can generate trajectories for unseen areas within the same city
    - *City-to-city generalization* (Section 4.3): Tests whether models can transfer knowledge between entirely different cities

The evaluation measures take on different significance when assessing generalization. KL(S∥R) becomes particularly important for out-of-distribution evaluation as it tests whether the synthetic distribution covers the full support of the real distribution in previously unseen regions. Meanwhile, KL(R∥S) reveals whether the model avoids generating implausible trajectories in new environments. The Pattern Score helps determine if important spatial areas in the new environment are captured, while TSTR directly measures the usefulness of the generated data for downstream tasks in these new settings.

Our experiments in Sections 4.3 demonstrate how this comprehensive evaluation identifies genuine generalization capabilities rather than mere memorization. The combination of measures reveals whether a model has learned transferable spatial dynamics or is simply reproducing patterns from its training data. This evaluation approach provides a more rigorous assessment of generalization than traditional machine learning benchmarks, which typically only test in-distribution generalization.

## F Additional Results

In Section 4 we present the results. Here we first provide additional visualization, focusing on more detailed view of and also providing a closer-look at the synthetic data generated. We also provide full tables as a complement to the tables provide figures that are averaged across datasets.

### F.1 Additional visualization

The visualizations presented in this section supports the quantitative results in Section 4.1 of the main paper. Specifically, Figures 6–8 demonstrate TDDM's ability to generate synthetic trajectories that closely match the spatial distribution patterns of the original data across all three datasets (Geolife, Porto, and Cabspotting).

These visualizations highlight two key aspects of TDDM's performance claimed in the main paper: (1) the high fidelity of individual generated trajectories, which follow road networks and maintain realistic movement patterns; and (2) the proportionality of the generated distribution, as shown by the similarity between the heatmaps of real and synthetic data. The comparison with baseline methods in Figures 2, 4 and 5 visually confirms TDDM's strong performance reported in Table 7.

The generalization capabilities of TDDM are further illustrated in Figures 12–14, which show how our model trained on a limited portion of each city can generate high-quality trajectories for previously unseen areas. Additionally, Figures 9–11 provide a closer look at trajectories in several regions

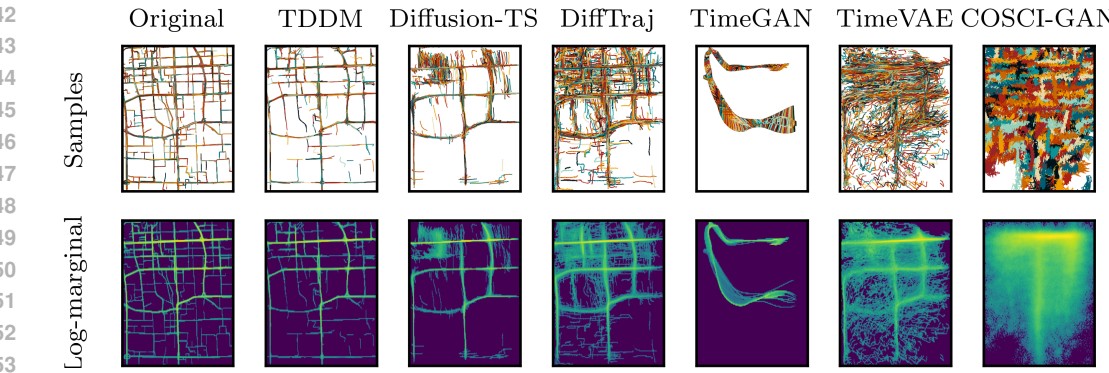

Figure 4: Comparison between original and synthetic trajectories for Geolife dataset. First row shows individual trajectory samples while bottom shows log-density heatmaps of all observations. The synthetic data of the proposed model, TDDM, (second column) most closely matches both the individual trajectory patterns and overall density distribution of the original data.

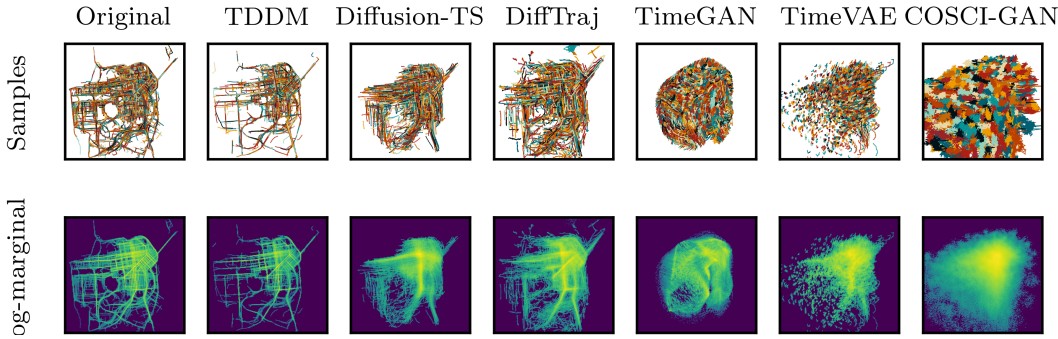

Figure 5: Comparison between original and synthetic trajectories for Cabspotting dataset. First row shows individual trajectory samples while bottom shows log-density heatmaps of all observations. The synthetic data of the proposed model, TDDM, (second column) most closely matches both the individual trajectory patterns and overall density distribution of the original data.

chosen at random, across the three datasets, demonstrating the consistent quality of our generated data compared to real trajectories and baseline methods.

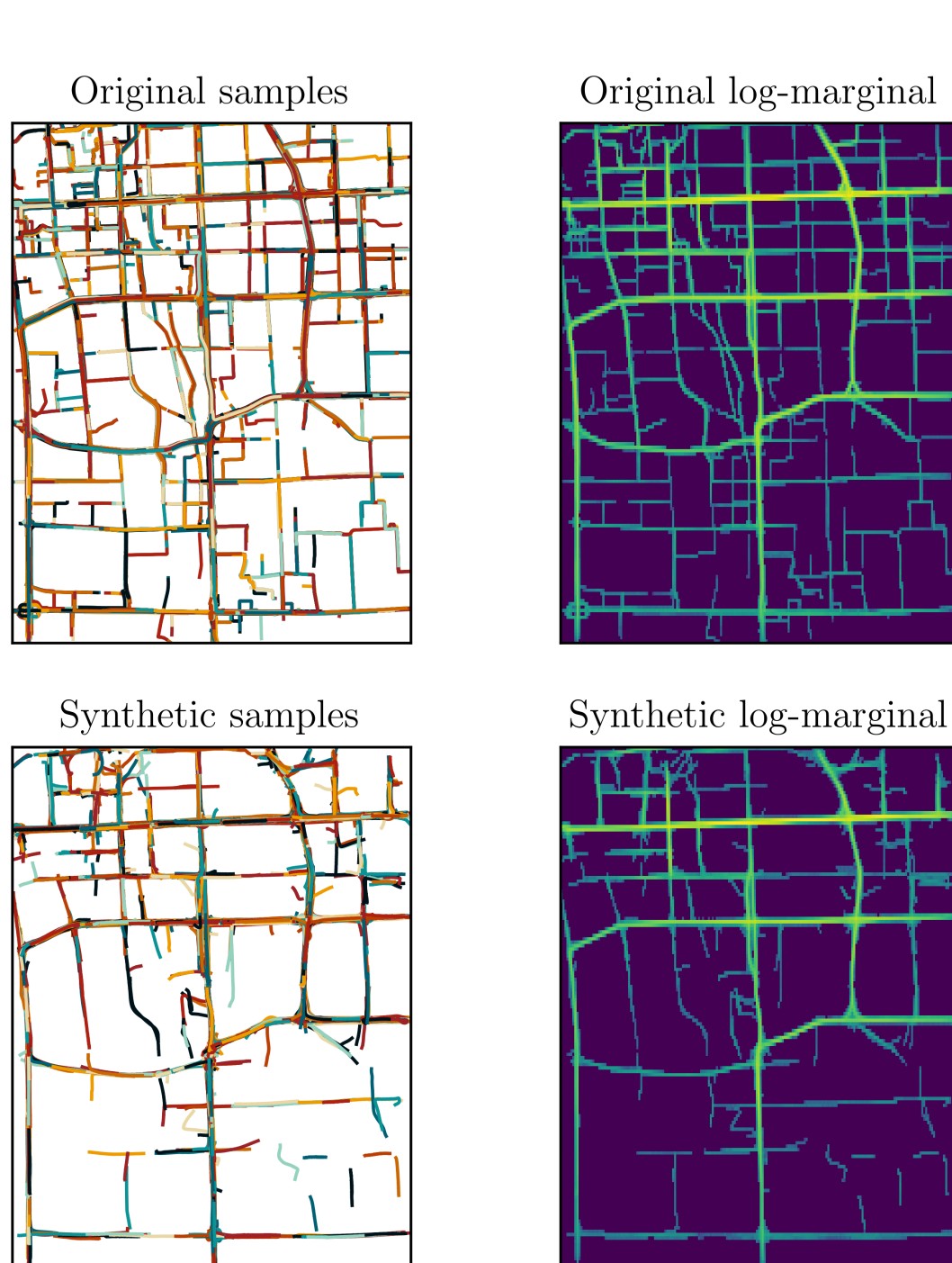

Figure 6: Detailed visualization for Geolife dataset. Comparison between training data (top) and synthetic trajectories from our method (bottom). Left panels show individual trajectory samples while right panels show log-density heatmaps of all points in the dataset. The synthetic samples closely match both the individual trajectory patterns and overall density distribution of the original data.

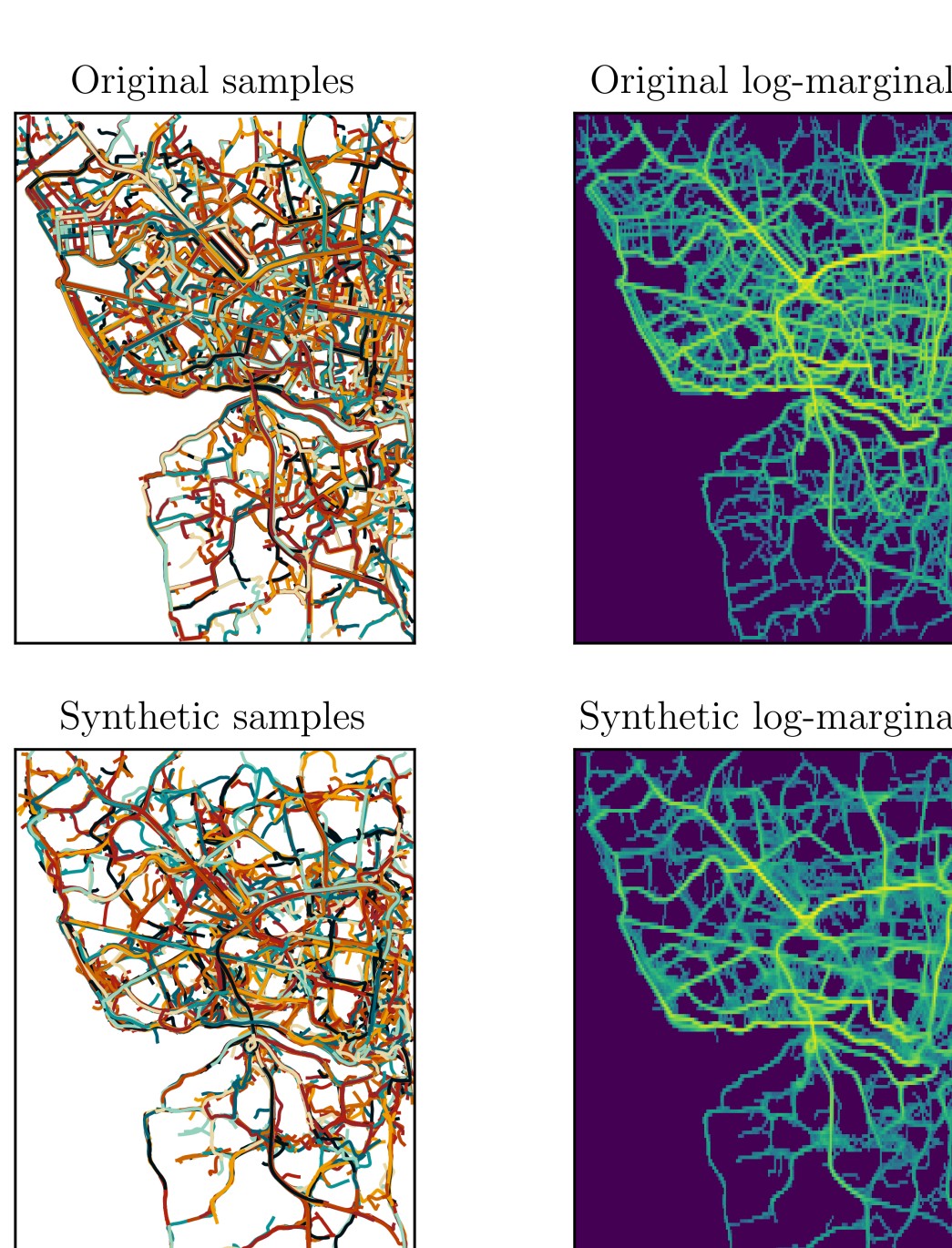

Figure 7: Detailed visualization for Porto dataset. Comparison between training data (top) and synthetic trajectories from our method (bottom). Left panels show individual trajectory samples while right panels show log-density heatmaps of all points in the dataset. The synthetic samples closely match both the individual trajectory patterns and overall density distribution of the original data.

Original samples 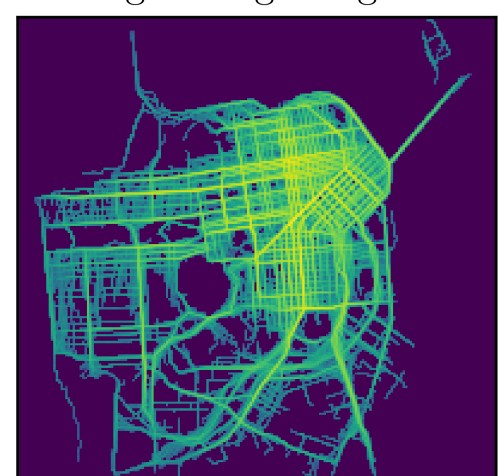

Original log-marginal

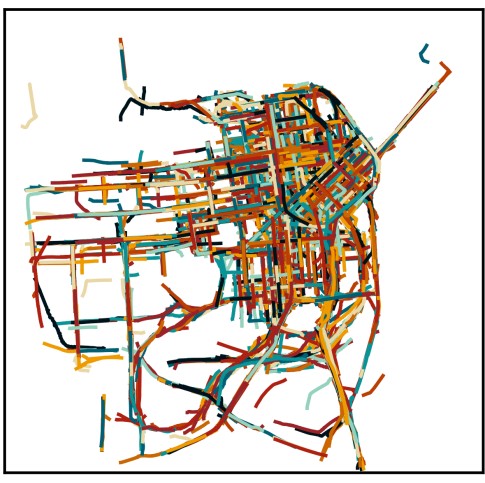 Synthetic samples

Synthetic log-marginal 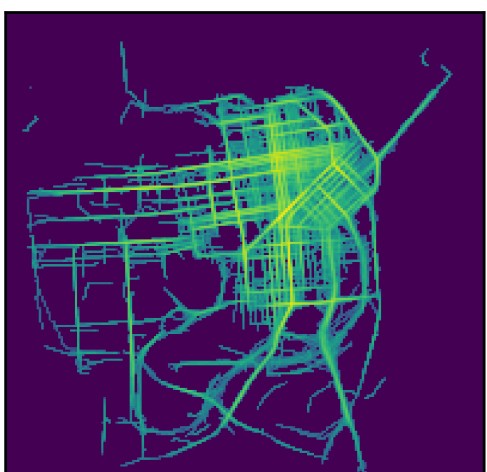

Figure 8: Detailed visualization for Cabspotting dataset. Comparison between training data (top) and synthetic trajectories from our method (bottom). Left panels show individual trajectory samples while right panels show log-density heatmaps of all points in the dataset. The synthetic samples closely match both the individual trajectory patterns and overall density distribution of the original data.

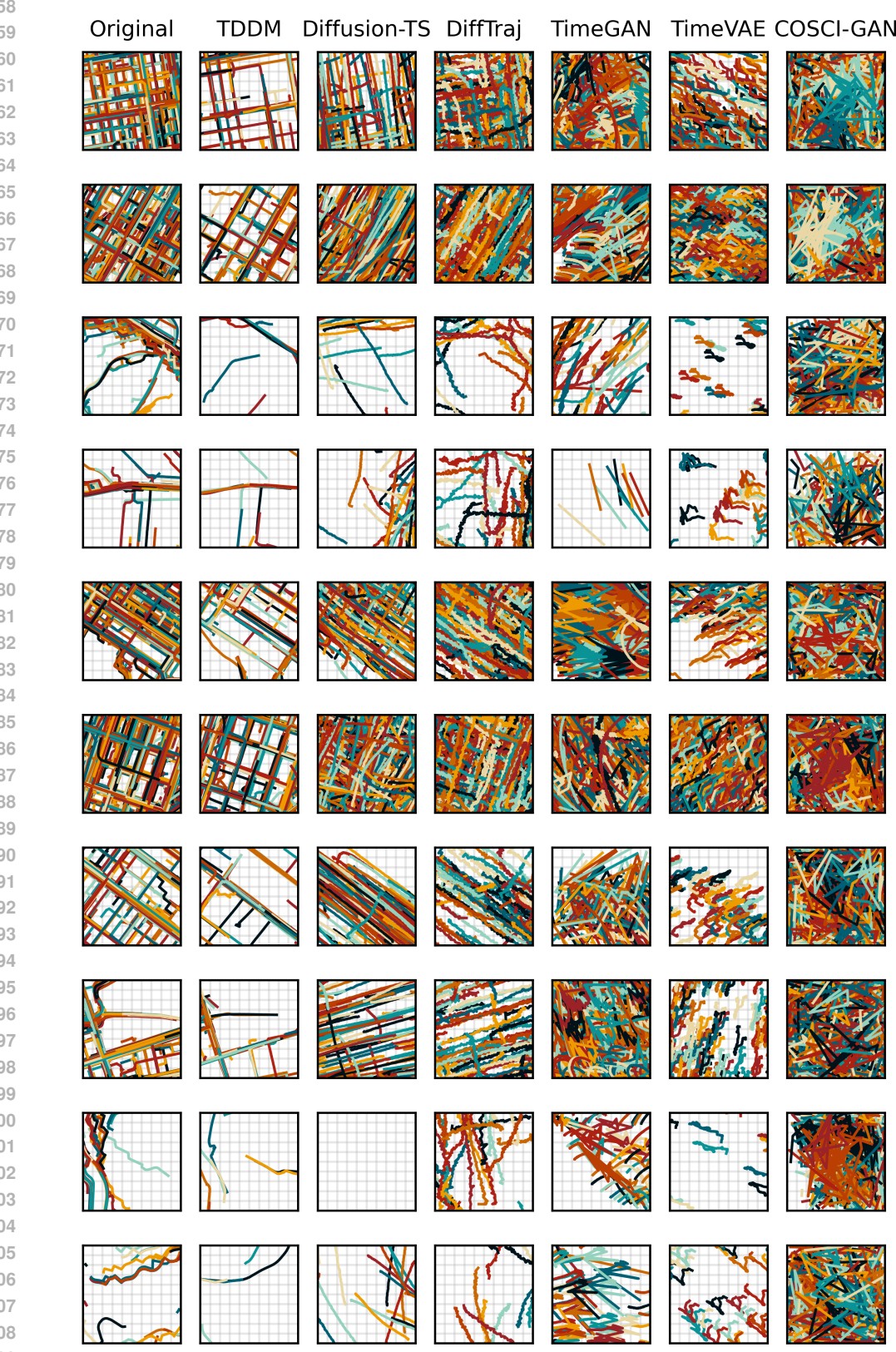

Figure 9: Random samples from training data and synthetic data across 11 different regions chosen at random, all from Cabspotting

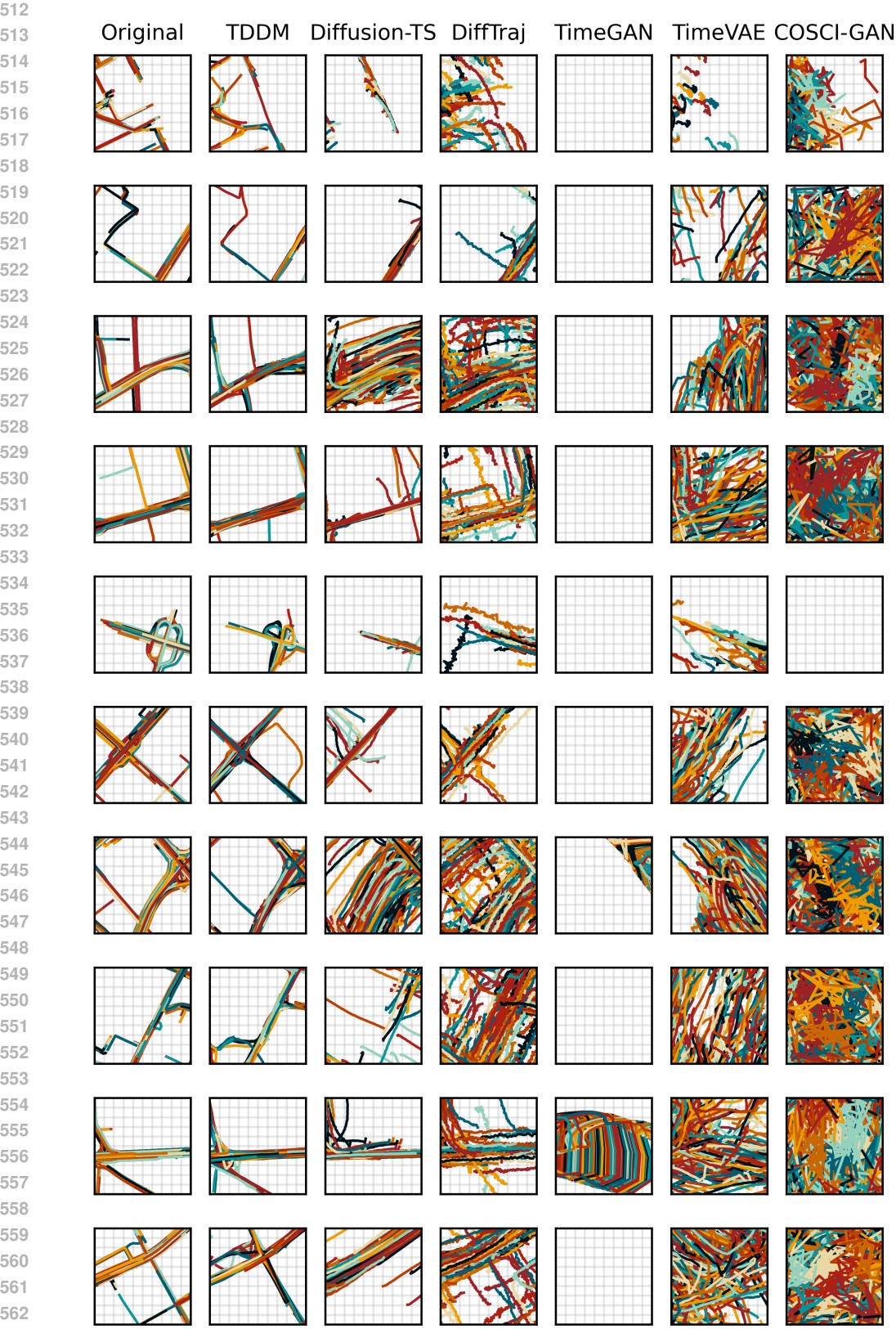

Figure 10: Random samples from training data and synthetic data across 11 different regions chosen at random, all from Geolife

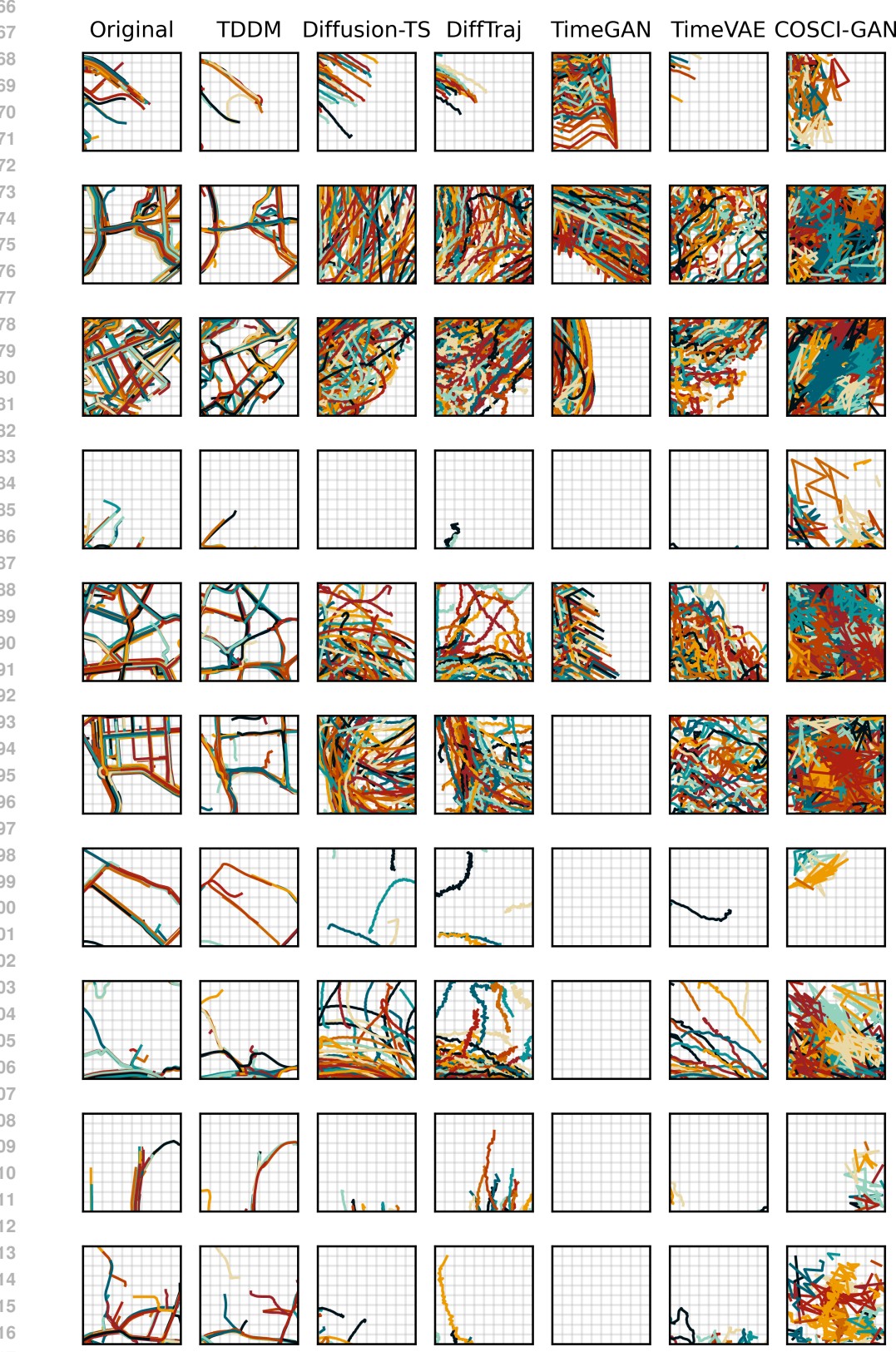

Figure 11: Random samples from training data and synthetic data across 10 different regions chosen at random, all from Porto.

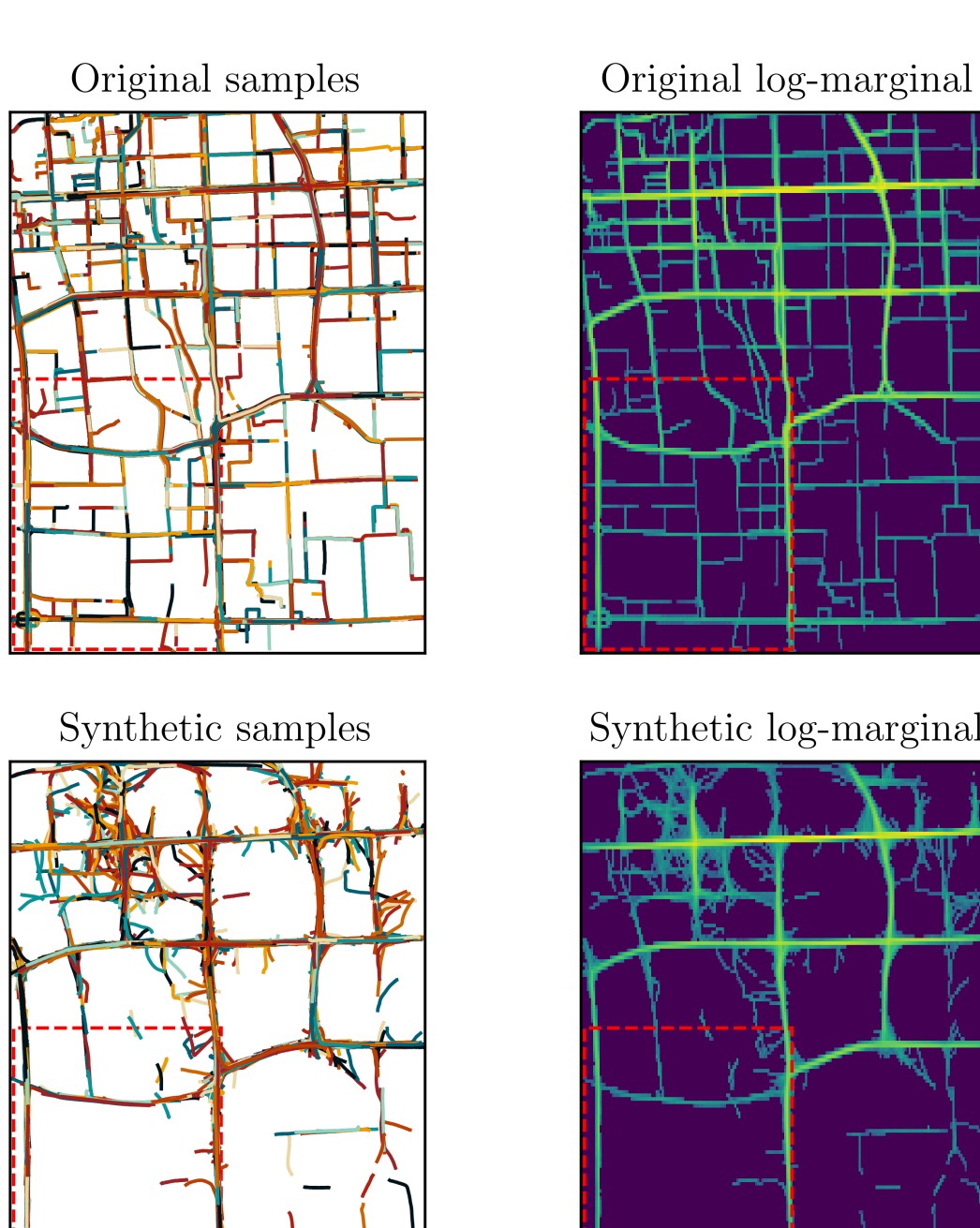

Figure 12: Generalization experiment. The model is trained on the lower left quadrant and used to generate data on the remaining geographical area. *Top left:* Data from Geolife, the lower-left quadrant of which used for training. *Top right:* heatmap of training data, *Bottom left:* synthetic trajectories. *Bottom right:* heatmap of the synthetic data.

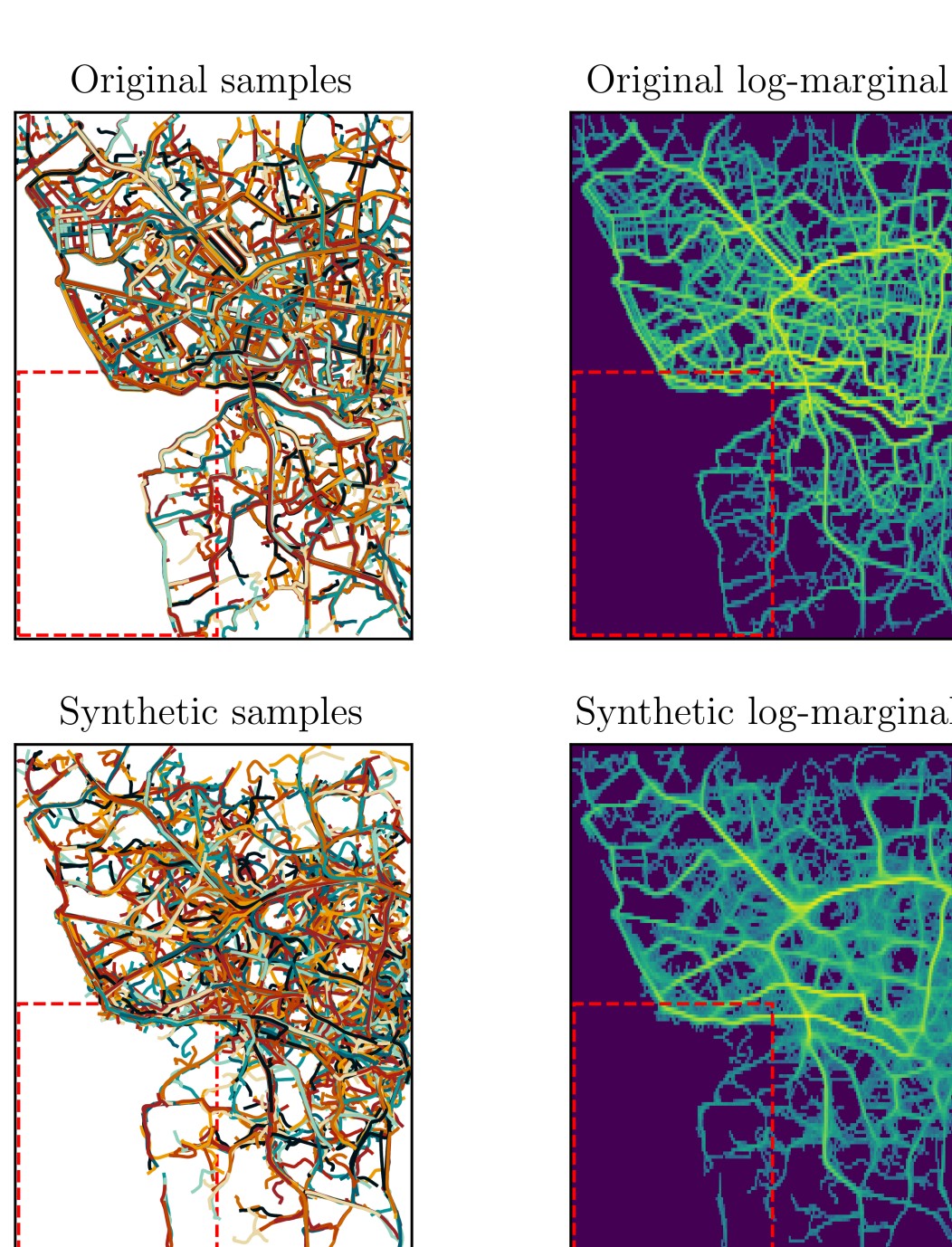

Figure 13: Generalization experiment. The model is trained on the top-left quadrant and used to generate data on the remaining geographical area. *Top left:* Data from Porto, the top-left quadrant of which used for training. *Top right:* heatmap of training data, *Bottom left:* synthetic trajectories. *Bottom right:* heatmap of the synthetic data.

### Original samples

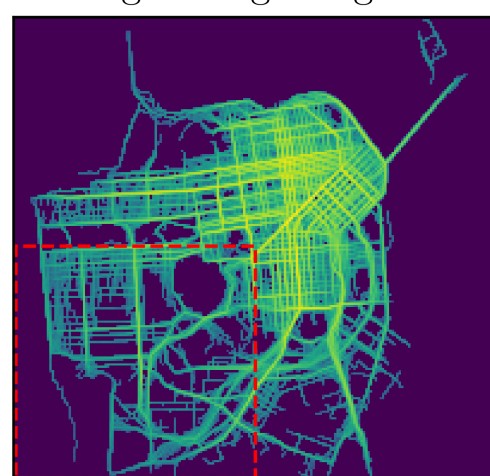

### Original log-marginal

### Synthetic samples

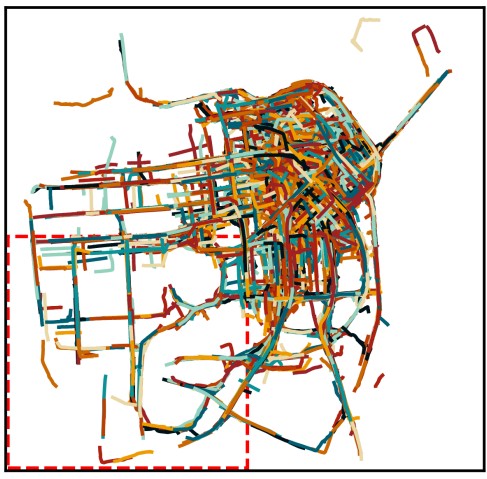

### Synthetic log-marginal

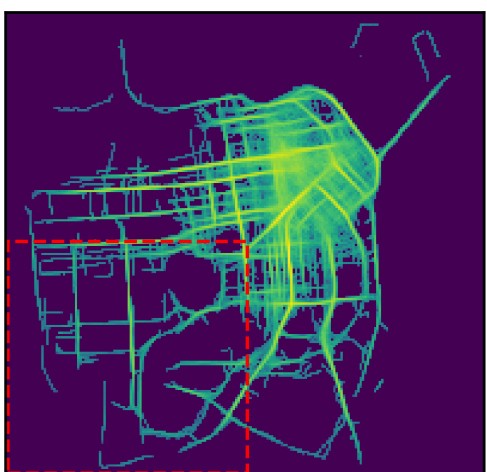

Figure 14: Generalization experiment. The model is trained on the lower left quadrant and used to generate data on the remaining geographical area. *Top left:* Data from Cabspotting, the lower-left quadrant of which used for training. *Top right:* heatmap of training data, *Bottom left:* synthetic trajectories. *Bottom right:* heatmap of the synthetic data.

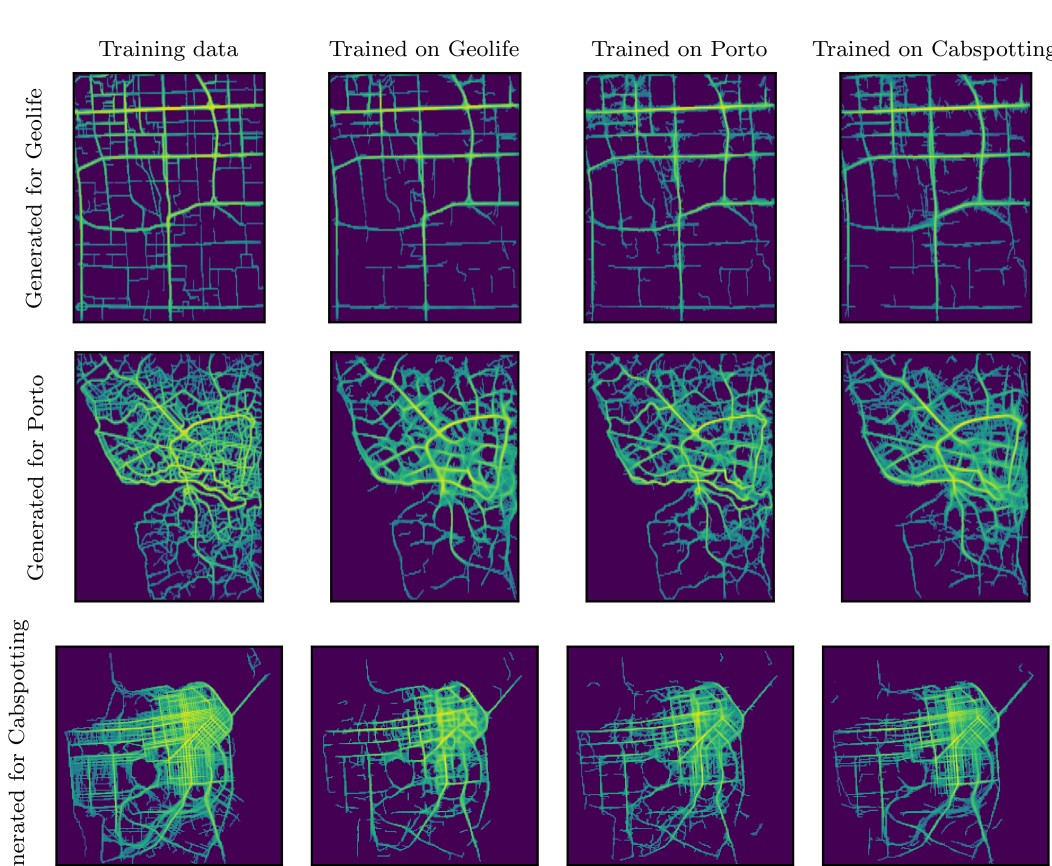

Figure 15: City-to-city generalization experiment visualization. The model is trained on one city and then used to generate data on the remaining geographical area using spatial priors.

## F.2 LENGTH DISTRIBUTION ANALYSIS

To provide deeper insight into the Length Error measure described in Section 4, we visualize the length distributions (distance between consecutive trajectory points) for both training and synthetic data across all experimental settings. Each histogram shows the normalized distribution of Euclidean distances between consecutive points, with the Jensen-Shannon divergence between these distributions reported in the top-right corner.

These visualizations demonstrate TDDM's ability to accurately match the fine-grained movement patterns of real trajectories. The close alignment between training (teal) and synthetic (red) distributions across diverse experimental conditions validates the quantitative Length Error results presented in the main paper.

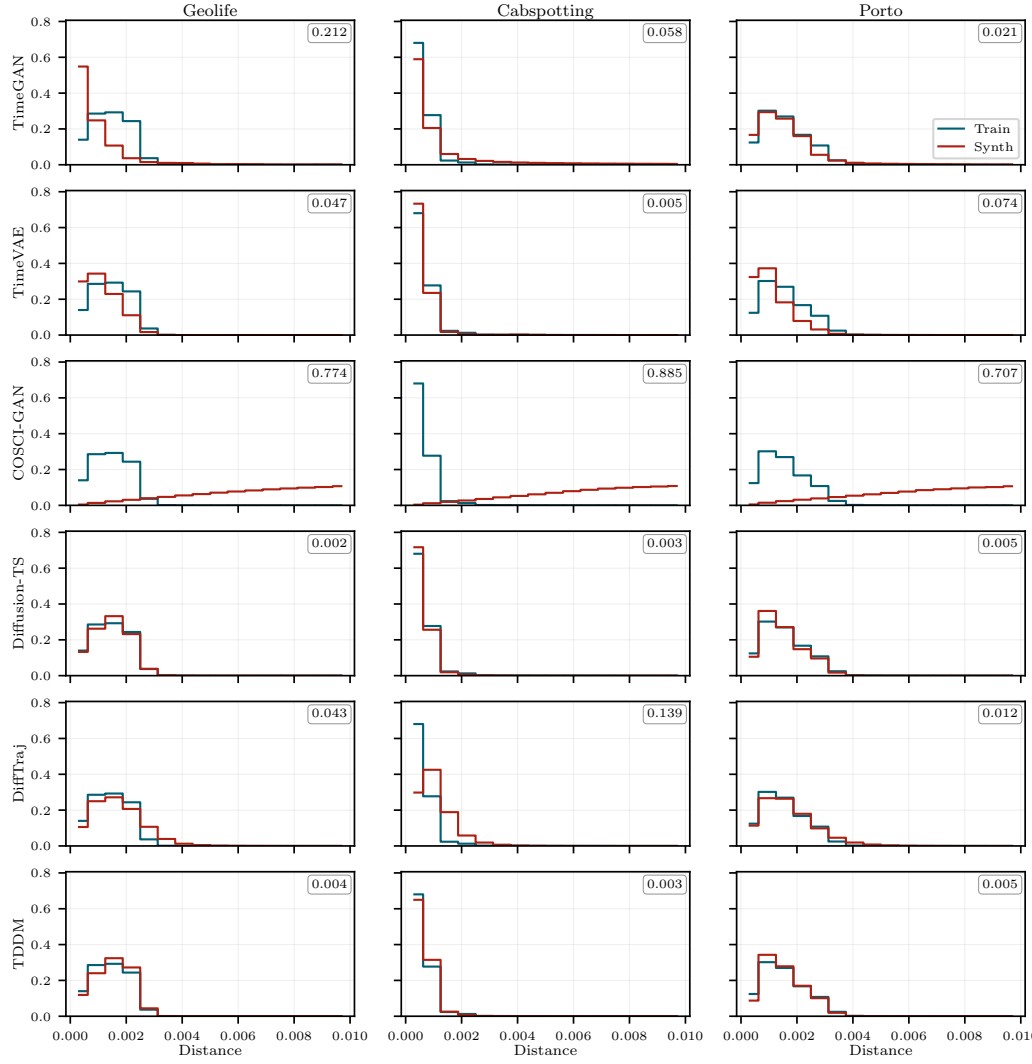

Figure 16: Length distribution comparison for unconditional generation benchmark. Each subplot shows the distribution of distances between consecutive trajectory points for training data (teal) and synthetic data (red) across three datasets (Geolife, Cabspotting, Porto) and six methods. The Jensen-Shannon divergence is shown in the top-right corner of each subplot.

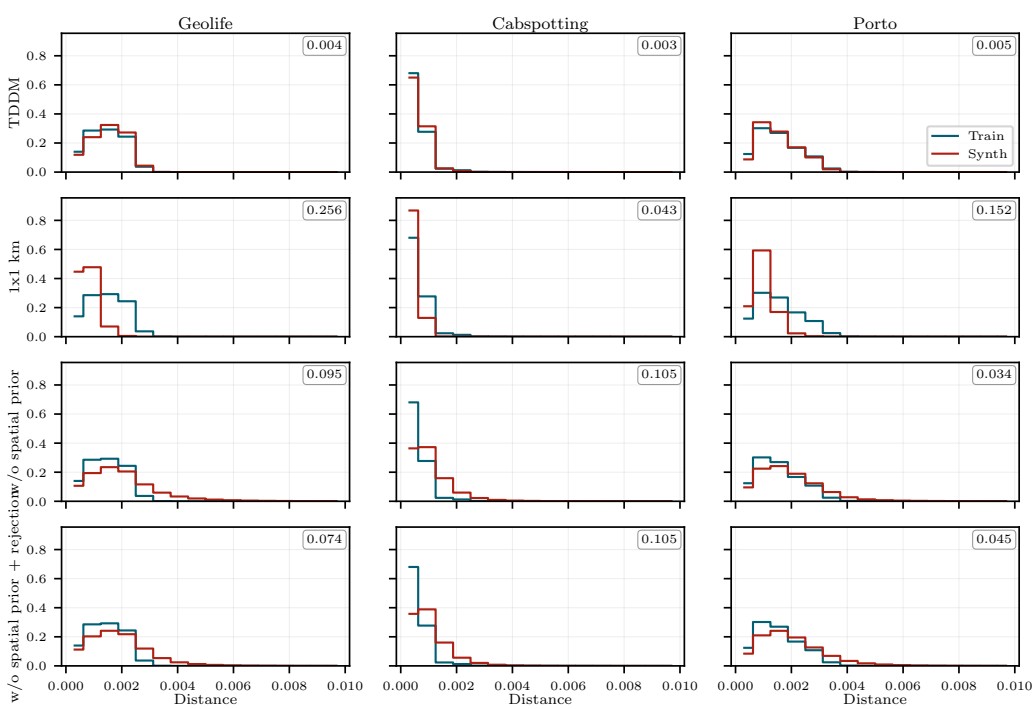

Figure 17: Length distribution comparison for ablation study. Shows the impact of different architectural choices on TDDM's ability to match the length distribution of training data.

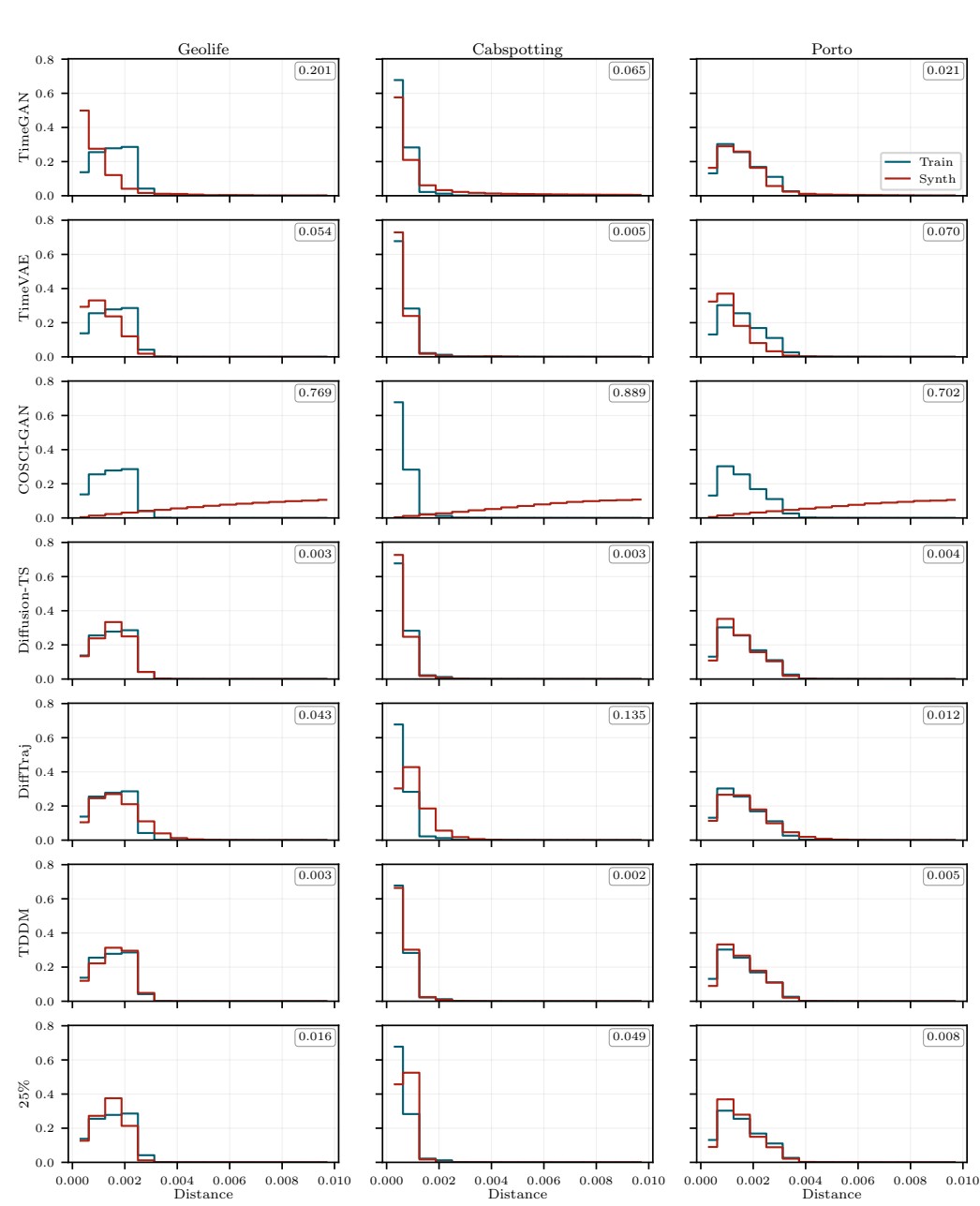

Figure 18: Length distribution comparison for intra-city generalization experiments. Models are trained on 25% of the city and evaluated on the full area.

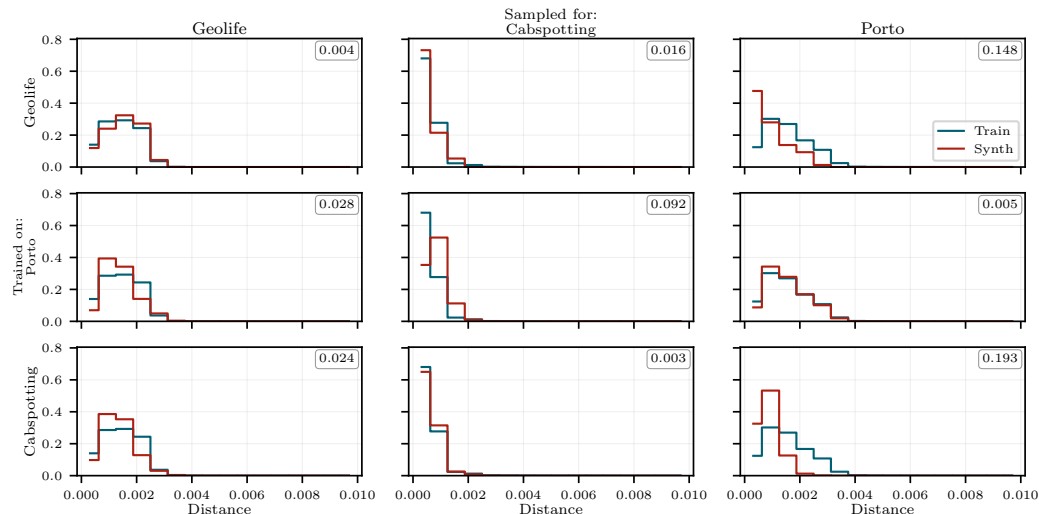

Figure 19: Length distribution comparison for city-to-city generalization experiments. Each row shows a model trained on one city and evaluated on all three cities.

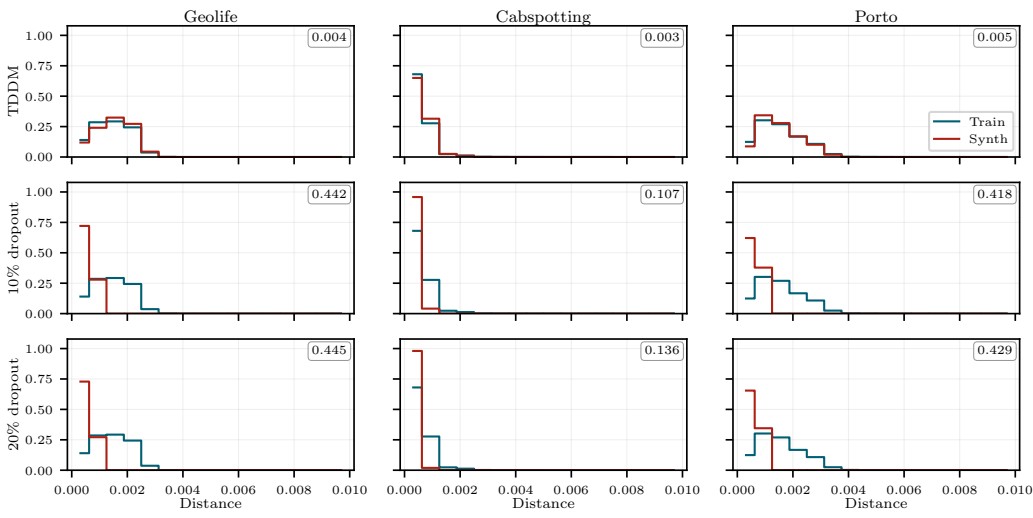

Figure 20: Length distribution comparison for map dropout robustness experiments. Shows TDDM's performance with varying levels of missing road network information.

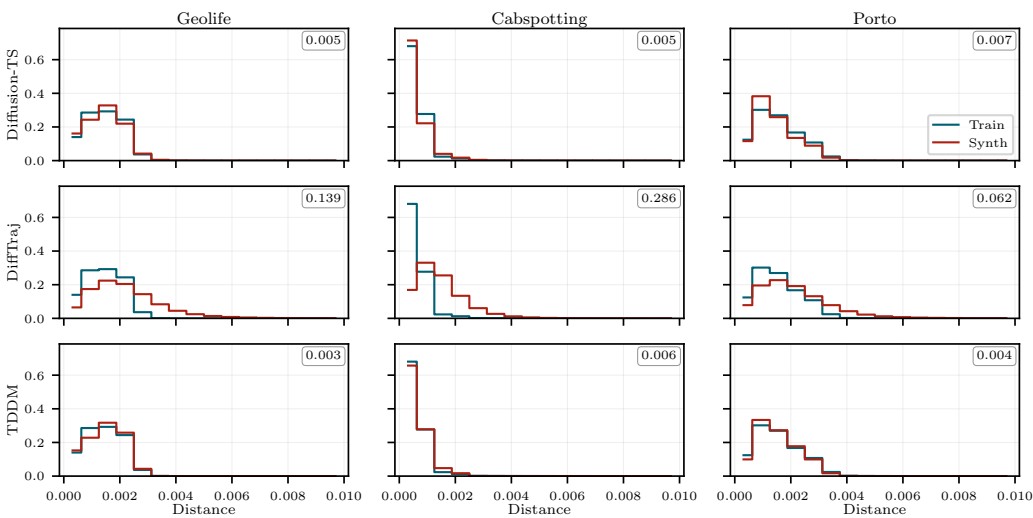

Figure 21: Length distribution comparison for models trained without map matching.

### F.3 QUANTITATIVE RESULTS

Complementing the summary tables, we list the tables with the quantitative results.

More specifically:

- *Large-scale unconditional trajectory generation* (4.1), Table 7.
- *Ablation study* (4.2), Table 8.
- *Intra-city* (4.3), Table 11.
- *City-to-city* (4.3), Table 12.
- *Map matching ablation*, Table 9.
- *Robustness to map dropout*, Table 10.

Table 7: Evaluation of different models' performance across several datasets and measures.

| Measure | Model | Geolife | Porto | Cabspotting |
|---|---|---|---|---|
| TSTR ($\downarrow$) | TimeGAN | $0.052 \pm 0.031$ | $0.038 \pm 0.025$ | $0.020 \pm 0.012$ |
| | TimeVAE | $0.017 \pm 0.010$ | $0.016 \pm 0.007$ | $0.021 \pm 0.011$ |
| | COSCI-GAN | $0.023 \pm 0.006$ | $0.021 \pm 0.005$ | $0.024 \pm 0.009$ |
| | Diffusion-TS | $\underline{0.013 \pm 0.009}$ | $\underline{0.010 \pm 0.005}$ | $0.021 \pm 0.008$ |
| | DiffTraj | $\underline{0.013 \pm 0.004}$ | $0.014 \pm 0.004$ | $\mathbf{0.013 \pm 0.007}$ |
| | TDDM | $\mathbf{0.010 \pm 0.005}$ | $\mathbf{0.009 \pm 0.004}$ | $\underline{0.015 \pm 0.008}$ |
| KL($S \parallel R$) ($\downarrow$) | TimeGAN | 5.566 | 3.606 | 1.933 |
| | TimeVAE | 2.786 | 2.688 | 1.615 |
| | COSCI-GAN | 4.650 | 2.910 | 1.578 |
| | Diffusion-TS | $\underline{0.730}$ | 2.114 | 1.340 |
| | DiffTraj | 1.498 | $\underline{2.025}$ | $\underline{1.259}$ |
| | TDDM | $\mathbf{0.202}$ | $\mathbf{0.437}$ | $\mathbf{0.265}$ |
| KL($R \parallel S$) ($\downarrow$) | TimeGAN | 3.823 | 2.632 | 1.302 |
| | TimeVAE | 1.236 | 1.445 | 1.124 |
| | COSCI-GAN | 2.453 | 1.696 | 1.070 |
| | Diffusion-TS | $\underline{0.598}$ | 1.135 | 0.999 |
| | DiffTraj | 0.767 | $\underline{1.023}$ | $\underline{0.817}$ |
| | TDDM | $\mathbf{0.177}$ | $\mathbf{0.336}$ | $\mathbf{0.246}$ |
| $KL_{sym}$ ($\downarrow$) | TimeGAN | 4.695 | 3.119 | 1.618 |
| | TimeVAE | 2.011 | 2.067 | 1.369 |
| | COSCI-GAN | 3.552 | 2.303 | 1.324 |
| | Diffusion-TS | $\underline{0.664}$ | 1.625 | 1.169 |
| | DiffTraj | 1.132 | $\underline{1.524}$ | $\underline{1.038}$ |
| | TDDM | $\mathbf{0.190}$ | $\mathbf{0.386}$ | $\mathbf{0.255}$ |
| JS ($\downarrow$) | TimeGAN | 0.489 | 0.430 | 0.271 |
| | TimeVAE | 0.294 | 0.327 | 0.241 |
| | COSCI-GAN | 0.486 | 0.364 | 0.240 |
| | Diffusion-TS | $\underline{0.114}$ | 0.264 | 0.216 |
| | DiffTraj | 0.181 | $\underline{0.252}$ | $\underline{0.195}$ |
| | TDDM | $\mathbf{0.040}$ | $\mathbf{0.079}$ | $\mathbf{0.056}$ |
| $KL_{speed}$ ($\downarrow$) | TimeGAN | 0.324 | 0.361 | 0.710 |
| | TimeVAE | 0.351 | 0.240 | 0.083 |
| | COSCI-GAN | 6.581 | 5.216 | 7.592 |
| | Diffusion-TS | 0.038 | $\mathbf{0.015}$ | $\underline{0.053}$ |
| | DiffTraj | $\underline{0.031}$ | 0.035 | 0.312 |
| | TDDM | $\mathbf{0.011}$ | $\underline{0.020}$ | $\mathbf{0.007}$ |
| Density ($\downarrow$) | TimeGAN | 0.433 | 0.252 | 0.088 |
| | TimeVAE | 0.063 | 0.028 | 0.038 |
| | COSCI-GAN | 0.264 | 0.078 | 0.060 |
| | Diffusion-TS | $\underline{0.035}$ | $\underline{0.021}$ | $\underline{0.031}$ |
| | DiffTraj | 0.046 | 0.023 | $\underline{0.031}$ |
| | TDDM | $\mathbf{0.023}$ | $\mathbf{0.019}$ | $\mathbf{0.014}$ |
| Trip ($\downarrow$) | TimeGAN | 0.510 | 0.346 | 0.115 |
| | TimeVAE | 0.078 | 0.049 | 0.040 |
| | COSCI-GAN | 0.295 | 0.096 | 0.082 |
| | Diffusion-TS | $\underline{0.049}$ | $\underline{0.037}$ | $\underline{0.036}$ |
| | DiffTraj | 0.052 | $\mathbf{0.035}$ | 0.037 |
| | TDDM | $\mathbf{0.039}$ | $\underline{0.037}$ | $\mathbf{0.018}$ |
| Length ($\downarrow$) | TimeGAN | 0.212 | 0.021 | 0.058 |
| | TimeVAE | 0.047 | 0.074 | $\underline{0.005}$ |
| | COSCI-GAN | 0.774 | 0.707 | 0.885 |
| | Diffusion-TS | $\mathbf{0.002}$ | $\mathbf{0.005}$ | $\mathbf{0.003}$ |
| | DiffTraj | 0.043 | $\underline{0.012}$ | 0.139 |
| | TDDM | $\underline{0.004}$ | $\mathbf{0.005}$ | $\mathbf{0.003}$ |
| Pattern ($\uparrow$) | TimeGAN | 0.490 | 0.730 | 0.810 |
| | TimeVAE | 0.780 | 0.890 | 0.850 |
| | COSCI-GAN | 0.650 | 0.830 | 0.830 |
| | Diffusion-TS | $\mathbf{0.920}$ | $\underline{0.900}$ | 0.900 |
| | DiffTraj | 0.890 | 0.880 | $\underline{0.910}$ |
| | TDDM | $\underline{0.910}$ | $\mathbf{0.920}$ | $\mathbf{0.920}$ |

Table 8: Ablation study of the effect of region size and spatial prior.

| Measure | Model | Geolife | Porto | Cabspotting |
|---|---|---|---|---|
| TSTR ($\downarrow$) | TDDM | $0.010 \pm 0.005$ | $0.009 \pm 0.004$ | $0.015 \pm 0.008$ |
| | 1x1 km | $0.030 \pm 0.014$ | $0.021 \pm 0.011$ | $0.022 \pm 0.011$ |
| | w/o spatial prior | $\mathbf{0.009 \pm 0.006}$ | $0.011 \pm 0.005$ | $\mathbf{0.012 \pm 0.007}$ |
| | w/o spatial prior + rejection | $0.021 \pm 0.009$ | $\mathbf{0.007 \pm 0.004}$ | $0.015 \pm 0.008$ |
| KL($S \parallel R$) ($\downarrow$) | TDDM | $\mathbf{0.202}$ | $\mathbf{0.437}$ | $0.265$ |
| | 1x1 km | $0.279$ | $0.552$ | $\mathbf{0.184}$ |
| | w/o spatial prior | $0.926$ | $2.349$ | $1.433$ |
| | w/o spatial prior + rejection | $1.343$ | $2.550$ | $1.881$ |
| KL($R \parallel S$) ($\downarrow$) | TDDM | $\mathbf{0.177}$ | $\mathbf{0.336}$ | $0.246$ |
| | 1x1 km | $0.206$ | $0.579$ | $\mathbf{0.168}$ |
| | w/o spatial prior | $0.858$ | $1.314$ | $1.123$ |
| | w/o spatial prior + rejection | $0.890$ | $1.478$ | $1.388$ |
| KL$_{\text{sym}}$ ($\downarrow$) | TDDM | $\mathbf{0.190}$ | $\mathbf{0.386}$ | $0.255$ |
| | 1x1 km | $0.243$ | $0.566$ | $\mathbf{0.176}$ |
| | w/o spatial prior | $0.892$ | $1.831$ | $1.278$ |
| | w/o spatial prior + rejection | $1.117$ | $2.014$ | $1.635$ |
| JS ($\downarrow$) | TDDM | $\mathbf{0.040}$ | $\mathbf{0.079}$ | $0.056$ |
| | 1x1 km | $0.054$ | $0.121$ | $\mathbf{0.038}$ |
| | w/o spatial prior | $0.156$ | $0.296$ | $0.231$ |
| | w/o spatial prior + rejection | $0.187$ | $0.324$ | $0.285$ |
| KL$_{\text{speed}}$ ($\downarrow$) | TDDM | $\mathbf{0.011}$ | $\mathbf{0.020}$ | $\mathbf{0.007}$ |
| | 1x1 km | $1.097$ | $0.528$ | $0.123$ |
| | w/o spatial prior | $0.478$ | $0.124$ | $0.368$ |
| | w/o spatial prior + rejection | $0.406$ | $0.215$ | $0.644$ |
| Density ($\downarrow$) | TDDM | $0.023$ | $\mathbf{0.019}$ | $0.014$ |
| | 1x1 km | $\mathbf{0.021}$ | $0.033$ | $\mathbf{0.011}$ |
| | w/o spatial prior | $0.099$ | $0.043$ | $0.060$ |
| | w/o spatial prior + rejection | $0.086$ | $0.046$ | $0.055$ |
| Trip ($\downarrow$) | TDDM | $\mathbf{0.039}$ | $\mathbf{0.037}$ | $\mathbf{0.018}$ |
| | 1x1 km | $0.049$ | $0.066$ | $\mathbf{0.018}$ |
| | w/o spatial prior | $0.099$ | $0.055$ | $0.068$ |
| | w/o spatial prior + rejection | $0.103$ | $0.070$ | $0.070$ |
| Length ($\downarrow$) | TDDM | $\mathbf{0.004}$ | $\mathbf{0.005}$ | $\mathbf{0.003}$ |
| | 1x1 km | $0.256$ | $0.152$ | $0.043$ |
| | w/o spatial prior | $0.095$ | $0.034$ | $0.105$ |
| | w/o spatial prior + rejection | $0.074$ | $0.045$ | $0.105$ |
| Pattern ($\uparrow$) | TDDM | $0.910$ | $\mathbf{0.920}$ | $0.920$ |
| | 1x1 km | $\mathbf{0.930}$ | $0.910$ | $\mathbf{0.950}$ |
| | w/o spatial prior | $0.860$ | $0.850$ | $0.790$ |
| | w/o spatial prior + rejection | $0.880$ | $0.870$ | $0.830$ |

Table 9: Map matching ablation study. Models trained without map matching during preprocessing.

| Measure | Model | Geolife | Porto | Cabspotting |
|---|---|---|---|---|
| TSTR ($\downarrow$) | Diffusion-TS | $\underline{0.015 \pm 0.006}$ | $0.012 \pm 0.005$ | $0.020 \pm 0.009$ |
| | DiffTraj | $0.019 \pm 0.005$ | $\mathbf{0.007 \pm 0.004}$ | $\underline{0.013 \pm 0.007}$ |
| | TDDM | $\mathbf{0.013 \pm 0.010}$ | $\underline{0.010 \pm 0.004}$ | $\mathbf{0.011 \pm 0.005}$ |
| KL($S \parallel R$) ($\downarrow$) | Diffusion-TS | $\underline{1.196}$ | $\underline{2.401}$ | $\mathbf{1.591}$ |
| | DiffTraj | $3.068$ | $2.799$ | $\underline{1.623}$ |
| | TDDM | $\mathbf{0.797}$ | $\mathbf{1.779}$ | $1.747$ |
| KL($R \parallel S$) ($\downarrow$) | Diffusion-TS | $\underline{1.038}$ | $\mathbf{1.309}$ | $\underline{1.263}$ |
| | DiffTraj | $1.387$ | $1.545$ | $\mathbf{1.036}$ |
| | TDDM | $\mathbf{0.739}$ | $\underline{1.451}$ | $1.469$ |
| KL$_{\text{sym}}$ ($\downarrow$) | Diffusion-TS | $\underline{1.117}$ | $\underline{1.855}$ | $\underline{1.427}$ |
| | DiffTraj | $2.227$ | $2.172$ | $\mathbf{1.329}$ |
| | TDDM | $\mathbf{0.768}$ | $\mathbf{1.615}$ | $1.608$ |
| JS ($\downarrow$) | Diffusion-TS | $\underline{0.182}$ | $\underline{0.298}$ | $\underline{0.249}$ |
| | DiffTraj | $0.319$ | $0.341$ | $\mathbf{0.238}$ |
| | TDDM | $\mathbf{0.138}$ | $\mathbf{0.259}$ | $0.264$ |
| KL$_{\text{speed}}$ ($\downarrow$) | Diffusion-TS | $\underline{0.058}$ | $\underline{0.018}$ | $\underline{0.042}$ |
| | DiffTraj | $0.381$ | $0.034$ | $0.416$ |
| | TDDM | $\mathbf{0.012}$ | $\mathbf{0.013}$ | $\mathbf{0.017}$ |
| Density ($\downarrow$) | Diffusion-TS | $\underline{0.051}$ | $\underline{0.025}$ | $0.055$ |
| | DiffTraj | $0.119$ | $0.078$ | $\underline{0.043}$ |
| | TDDM | $\mathbf{0.038}$ | $\mathbf{0.022}$ | $\mathbf{0.040}$ |
| Trip ($\downarrow$) | Diffusion-TS | $\underline{0.064}$ | $\underline{0.044}$ | $0.065$ |
| | DiffTraj | $0.127$ | $0.087$ | $\mathbf{0.051}$ |
| | TDDM | $\mathbf{0.052}$ | $\mathbf{0.037}$ | $\underline{0.057}$ |
| Length ($\downarrow$) | Diffusion-TS | $\underline{0.005}$ | $\underline{0.007}$ | $\mathbf{0.005}$ |
| | DiffTraj | $0.139$ | $0.062$ | $0.286$ |
| | TDDM | $\mathbf{0.003}$ | $\mathbf{0.004}$ | $\underline{0.006}$ |
| Pattern ($\uparrow$) | Diffusion-TS | $\underline{0.870}$ | $\mathbf{0.890}$ | $0.790$ |
| | DiffTraj | $0.800$ | $0.810$ | $\underline{0.860}$ |
| | TDDM | $\mathbf{0.910}$ | $\underline{0.880}$ | $\mathbf{0.880}$ |

Table 10: Robustness to map dropout. Models trained with random dropout of map information during training.

| Measure | Model | Geolife | Porto | Cabspotting |
|---|---|---|---|---|
| TSTR ($\downarrow$) | TDDM | **0.010 ± 0.005** | **0.009 ± 0.004** | **0.015 ± 0.008** |
| | 10% dropout | 0.027 ± 0.014 | 0.030 ± 0.016 | 0.020 ± 0.011 |
| | 20% dropout | 0.025 ± 0.011 | 0.033 ± 0.017 | 0.021 ± 0.010 |
| KL($S \parallel R$) ($\downarrow$) | TDDM | **0.202** | 0.437 | 0.265 |
| | 10% dropout | 0.246 | 0.213 | **0.214** |
| | 20% dropout | 0.243 | **0.211** | 0.215 |
| KL($R \parallel S$) ($\downarrow$) | TDDM | **0.177** | 0.336 | **0.246** |
| | 10% dropout | 0.379 | 0.200 | 0.253 |
| | 20% dropout | 0.376 | **0.197** | 0.256 |
| KL$_{\mathrm{sym}}$ ($\downarrow$) | TDDM | **0.190** | 0.386 | 0.255 |
| | 10% dropout | 0.313 | 0.207 | **0.234** |
| | 20% dropout | 0.310 | **0.204** | 0.235 |
| JS ($\downarrow$) | TDDM | **0.040** | 0.079 | 0.056 |
| | 10% dropout | 0.061 | 0.045 | **0.049** |
| | 20% dropout | 0.061 | **0.044** | **0.049** |
| KL$_{\mathrm{speed}}$ ($\downarrow$) | TDDM | **0.011** | **0.020** | **0.007** |
| | 10% dropout | 4.442 | 2.422 | 0.837 |
| | 20% dropout | 4.523 | 3.096 | 1.279 |
| Density ($\downarrow$) | TDDM | 0.023 | 0.019 | 0.014 |
| | 10% dropout | **0.016** | **0.011** | **0.011** |
| | 20% dropout | **0.016** | **0.011** | **0.011** |
| Trip ($\downarrow$) | TDDM | 0.039 | 0.037 | 0.018 |
| | 10% dropout | **0.032** | **0.034** | **0.017** |
| | 20% dropout | 0.033 | 0.035 | 0.018 |
| Length ($\downarrow$) | TDDM | **0.004** | **0.005** | **0.003** |
| | 10% dropout | 0.442 | 0.418 | 0.107 |
| | 20% dropout | 0.445 | 0.429 | 0.136 |
| Pattern ($\uparrow$) | TDDM | 0.910 | 0.920 | 0.920 |
| | 10% dropout | **0.920** | **0.930** | **0.950** |
| | 20% dropout | 0.910 | 0.920 | **0.950** |

Table 11: Generalization experiment.

| Measure | Trained on | Geolife | Porto | Cabspotting |
|---|---|---|---|---|
| TSTR ($\downarrow$) | 100% | $\mathbf{0.007 \pm 0.004}$ | $\mathbf{0.006 \pm 0.003}$ | $\underline{0.016 \pm 0.009}$ |
|  | 25% | $\underline{0.011 \pm 0.006}$ | $\underline{0.010 \pm 0.004}$ | $\mathbf{0.014 \pm 0.008}$ |
| KL($S \parallel R$) ($\downarrow$) | 100% | $\mathbf{0.199}$ | $\mathbf{0.448}$ | $\mathbf{0.266}$ |
|  | 25% | $\underline{0.424}$ | $\underline{0.916}$ | $\underline{0.461}$ |
| KL($R \parallel S$) ($\downarrow$) | 100% | $\mathbf{0.168}$ | $\mathbf{0.340}$ | $\mathbf{0.245}$ |
|  | 25% | $\underline{0.373}$ | $\underline{0.629}$ | $\underline{0.403}$ |
| KL$_{\text{sym}}$ ($\downarrow$) | 100% | $\mathbf{0.184}$ | $\mathbf{0.394}$ | $\mathbf{0.256}$ |
|  | 25% | $\underline{0.399}$ | $\underline{0.773}$ | $\underline{0.432}$ |
| JS ($\downarrow$) | 100% | $\mathbf{0.039}$ | $\mathbf{0.080}$ | $\mathbf{0.056}$ |
|  | 25% | $\underline{0.076}$ | $\underline{0.145}$ | $\underline{0.093}$ |
| KL$_{\text{speed}}$ ($\downarrow$) | 100% | $\mathbf{0.011}$ | $\mathbf{0.019}$ | $\mathbf{0.007}$ |
|  | 25% | $\underline{0.046}$ | $\underline{0.027}$ | $\underline{0.230}$ |
| Density ($\downarrow$) | 100% | $\mathbf{0.019}$ | $\mathbf{0.016}$ | $\mathbf{0.011}$ |
|  | 25% | $\underline{0.025}$ | $\underline{0.020}$ | $\underline{0.014}$ |
| Trip ($\downarrow$) | 100% | $\mathbf{0.032}$ | $\mathbf{0.037}$ | $\mathbf{0.013}$ |
|  | 25% | $\underline{0.039}$ | $\underline{0.048}$ | $\underline{0.019}$ |
| Length ($\downarrow$) | 100% | $\mathbf{0.003}$ | $\mathbf{0.005}$ | $\mathbf{0.002}$ |
|  | 25% | $\underline{0.016}$ | $\underline{0.008}$ | $\underline{0.049}$ |
| Pattern ($\uparrow$) | 100% | $\mathbf{0.930}$ | $\mathbf{0.930}$ | $\mathbf{0.960}$ |
|  | 25% | $\underline{0.910}$ | $\underline{0.900}$ | $\underline{0.950}$ |

Table 12: City to city generalization experiment. The model is trained on one city and then is sampled for a target city using the marginal distribution.

| Measure | Trained on | Sampled for | | |
|---|---|---|---|---|
| | | Geolife | Porto | Cabspotting |
| TSTR ($\downarrow$) | Geolife | **0.010 ± 0.005** | 0.015 ± 0.007 | 0.017 ± 0.008 |
| | Porto | 0.011 ± 0.006 | **0.009 ± 0.004** | **0.009 ± 0.003** |
| | Cabspotting | 0.012 ± 0.006 | 0.010 ± 0.005 | 0.015 ± 0.008 |
| KL($S \parallel R$) ($\downarrow$) | Geolife | **0.202** | 1.168 | 0.637 |
| | Porto | 0.294 | **0.437** | 0.421 |
| | Cabspotting | 0.313 | 0.907 | **0.265** |
| KL($R \parallel S$) ($\downarrow$) | Geolife | **0.177** | 0.796 | 0.579 |
| | Porto | 0.232 | **0.336** | 0.394 |
| | Cabspotting | 0.258 | 0.641 | **0.246** |
| KL$_{\text{sym}}$ ($\downarrow$) | Geolife | **0.190** | 0.982 | 0.608 |
| | Porto | 0.263 | **0.386** | 0.407 |
| | Cabspotting | 0.286 | 0.774 | **0.255** |
| JS ($\downarrow$) | Geolife | **0.040** | 0.175 | 0.123 |
| | Porto | 0.054 | **0.079** | 0.087 |
| | Cabspotting | 0.058 | 0.145 | **0.056** |
| KL$_{\text{speed}}$ ($\downarrow$) | Geolife | **0.011** | 0.505 | 0.139 |
| | Porto | 0.097 | **0.020** | 0.380 |
| | Cabspotting | 0.083 | 0.703 | **0.007** |
| Density ($\downarrow$) | Geolife | 0.023 | 0.021 | 0.016 |
| | Porto | **0.021** | 0.019 | 0.015 |
| | Cabspotting | 0.026 | 0.019 | **0.014** |
| Trip ($\downarrow$) | Geolife | **0.039** | 0.040 | 0.022 |
| | Porto | 0.047 | **0.037** | 0.026 |
| | Cabspotting | 0.042 | 0.042 | **0.018** |
| Length ($\downarrow$) | Geolife | **0.004** | 0.148 | 0.016 |
| | Porto | 0.028 | **0.005** | 0.092 |
| | Cabspotting | 0.024 | 0.193 | **0.003** |
| Pattern ($\uparrow$) | Geolife | 0.910 | **0.920** | 0.930 |
| | Porto | **0.930** | 0.920 | 0.930 |
| | Cabspotting | 0.920 | 0.910 | 0.920 |

