# OpenReview forum: "Learning to Deaggregate: Large-scale Trajectory Generation with Spatial Priors"
_ICLR.cc/2026/Conference — Submitted to ICLR 2026_

### Official Review · Reviewer_YR6q · 2025-10-26

**Soundness:** 3
**Presentation:** 2
**Contribution:** 3
**Rating:** 6
**Confidence:** 3

**Summary:**

This paper proposes TDDM (Temporal Deaggregation Diffusion Model), a novel diffusion-based framework for large-scale, unconditional, and generalizable trajectory generation.

Instead of directly modeling trajectories, TDDM decouples the task into: 1) a spatial occupancy prior H, representing aggregated mobility intensity, and 2) a temporal diffusion model, which learns local motion dynamics in normalized regional coordinates.

A canonicalization module performs similarity transformations (translation–rotation–scaling) to map diverse city regions into a shared local coordinate frame, allowing parameter reuse and cross-city generalization.
The spatial prior is represented as a marginal heatmap token sequence conditioning the diffusion model, enabling spatial control while reducing overfitting risks associated with individual trajectory conditioning.

Experiments on three real-world GPS datasets (Beijing, Porto, San Francisco) show consistent improvements over GAN-, VAE-, and prior diffusion-based baselines in distribution alignment (KL, JS), fidelity (Pattern score), and generalization (TSTR).

The paper includes extensive evaluations, ablation studies, and ethical considerations, and commits to code/data release.

**Strengths:**

1.	Originality:
This paper introduces a new decomposition paradigm for trajectory generation (spatial prior + temporal diffusion). The idea of canonicalized spatial conditioning is practically useful for generalization.

2.	Technical Quality:
The probabilistic formulation is solid; diffusion-based denoising is well implemented. Ablations and quantitative results support the core claims.

3.	Significance:
This paper addresses a key challenge in human mobility modeling: privacy-constrained data scarcity and cross-domain transfer. TDDM consistently outperforms competing methods across three geographically diverse cities and multiple fidelity metrics, highlighting its strong potential for large-scale synthetic data generation and cross-domain transfer applications.

4.	Clarity:
Logical flow and clean architecture diagrams aid understanding; mathematical expressions are readable. Well organized supplementary materials include datasets, training details, and comparisons.

**Weaknesses:**

1.	Intra-city generalization result unexplained:
The near-identical results between 25% and 100% training coverage raise concerns of metric insensitivity or sampling bias. The paper should clarify region selection and provide repeated trials or visualizations.

2.	Lack of physical validity evaluation:
Generated trajectories are not checked against road networks or obstacles. Thus, “realism” is only statistical, not geometric.

3.	Ablation incompleteness:
Canonicalization ablation is coarse (all-or-nothing). Intermediate analyses (e.g., removing only rotation/scale) could strengthen causal claims.

**Questions:**

1.	How is the 25% subset of intra-city regions selected? Randomly or by spatial clustering? Were multiple subsets tested to ensure robustness?

2.	How sensitive are results to the spatial prior resolution (grid size of H)?

3.	Cross-city results are averaged, but the detailed results should be added.

4.	What about the analysis of failure cases (e.g., Porto→SF length error 0.193)?

5.	Could the authors provide a temporal similarity metric (e.g., DTW, speed KL) to complement spatial fidelity?

6.	The paper uses 8-layer Transformers with 500 diffusion steps. As city coverage grows, token count may scale quadratically. Please report per-trajectory inference time (GPU ms) and peak memory.

7.	Would TDDM generalize to cities with different traffic rules (e.g., left-hand driving in London/Tokyo)?

8.	Would integrating road network constraints (e.g., via map-based priors) further improve trajectory realism?

9.	Would integrating more realistic temporal features (e.g., speed, POI) further improve trajectory realism?

---

> ### Author Response · Authors · 2025-11-26
>
> We are grateful for the reviewer's thorough evaluation and constructive feedback, which we address below.
>
> **W1:** They are not. Although still outperforming the baselines, per-city metrics (Table 8) show TDDM-25% is 10–50% worse than TDDM-100% across nearly all metrics except Geolife Pattern. The model remains strong, but the performance gap is real and reflects reduced coverage and temporal fidelity in low-diversity settings. The aggregated numbers in Table 3 obscure this variation. We will clarify these results in the coming revision.
>
> **W2:** We introduced a map-matching evaluation comparing synthetic vs. map-matched trajectories, and include the training data for reference:
>
> | Model         | Geolife          | Porto            | Cabspotting      |
> | ------------- | ---------------- | ---------------- | ---------------- |
> | TimeGAN       | 23.6 (52.8%) | 22.9 (28.9%)     | 22.3 (39.0%) |
> | TimeVAE       | 25.0 (40.0%)     | 23.1 (25.1%)     | 22.7 (33.0%)     |
> | COSCI-GAN     | 24.2 (59.8%)     | 23.6 (22.7%)     | 22.6 (40.8%)     |
> | Diffusion-TS  | 25.5 (54.3%)     | 22.8 (24.1%) | 22.4 (31.8%)     |
> | DiffTraj      | 25.6 (55.4%)     | 23.3 (23.8%)     | 23.2 (14.6%)     |
> | TDDM          | 26.8 (42.1%)     | 22.8 (22.3%)     | 23.3 (21.0%)     |
> | **Training Data** | **26.9 (42.9%)**     | **22.7 (22.3%)** | **22.7 (13.4%)** |
>
> TDDM matches training-data behavior closely on Geolife and Porto; Cabspotting is modestly higher but comparable. Baselines that report lower distances typically incur higher failure rates, consistent with mode collapse toward easily-matched paths.
>
> **W3:** We agree finer-grained ablations (scale-only, rotation-only) would be informative and will note this as future work.
>
> **Q1:** We train on the bottom-left quadrant of each city (Figures 12–14), ensuring a consistent and geographically distinct training region across datasets.
>
> **Q2:** We use a balanced $64 \times 64$ grid over $3 \times 3$ km regions. Finer grids increase quadratic token cost. Coarser resolution reduces spatial information, which would likely degrade trajectory quality—particularly for capturing detailed road structure. However, excessively fine resolution may not provide proportional benefits given GPS measurement precision.
>
> **Q3:** Full city-to-city matrices are provided in Appendix E.2 (Table 9). We kept averages in the main text for space reasons.
>
> **Q4:** Porto has a heavier-tailed length distribution than SF. TDDM trained on SF cannot infer Porto’s tail behavior from H alone, generating shorter trajectories instead. We will include distribution plots in the revision.
>
> **Q5:** We implemented speed KL-divergence as a temporal fidelity metric, computed as the symmetric KL-divergence between velocity distributions of training and synthetic trajectories using 100-bin Bayesian-smoothed histograms:
>
> | Model        | Geolife   | Porto     | Cabspotting |
> | ------------ | --------- | --------- | ----------- |
> | TimeGAN      | 0.324     | 0.361     | 0.710       |
> | TimeVAE      | 0.351     | 0.240     | 0.083       |
> | COSCI-GAN    | 6.581     | 5.216     | 7.592       |
> | Diffusion-TS | 0.038     | **0.015** | _0.053_       |
> | DiffTraj     | _0.031_     | 0.035     | 0.312       |
> | TDDM         | **0.011** | _0.019_     | **0.007**   |
>
> TDDM achieves the lowest speed KL-divergence on two of three datasets (Geolife: 0.011, Cabspotting: 0.007), demonstrating strong temporal fidelity. On Porto, TDDM achieves competitive speed KL (0.019 vs. 0.015 for Diffusion-TS).
>
> **Q6:** TDDM operates on fixed $3 \times 3$ km regions, token count is constant across cities and sampling scales *linearly* with area. Per-trajectory inference on RTX 3090 Ti and per-batch memory requirements:
>
> | Batch size | Time per trajectory (ms) | Peak GPU memory (MB) |
> | ---------- | ------------------------ | -------------------- |
> | 1          | 1162                     | 313                  |
> | 8          | 427                      | 340                  |
> | 64         | 349                      | 558                  |
> | 256        | 340                      | 1307                 |
>
>
> **Q7:** The current TDDM cannot infer traffic directionality from the spatial prior H alone, as H contains only marginal occupancy information without directional cues. Training on right-hand driving cities and generalizing to left-hand driving cities would generate trajectories following right-hand patterns.
>
> A potential approximation would be reversing the temporal ordering of generated trajectories to flip traffic flow, though this would not perfectly capture left-hand dynamics since acceleration/deceleration/turning patterns may not be time-symmetric.
>
> Augmenting H with directional priors is a natural extension for future work.
>
> **Q8:** We believe so. Road hierarchy should help encode local speed structure missing from H. We intentionally avoid map dependency in this work for generality, but incorporating such priors is a promising extension.

---

> > ### Author Response · Authors · 2025-11-26
> > **Part 2**
> >
> > **Q9:** City-specific temporal signatures cannot be inferred from H alone. Extending H to include temporal marginals or contextual features would likely improve cross-city temporal fidelity while preserving zero-shot spatial generalization. We will note this as future work.

---

### Official Review · Reviewer_L1Qa · 2025-10-28

**Soundness:** 3
**Presentation:** 3
**Contribution:** 2
**Rating:** 4
**Confidence:** 3

**Summary:**

The paper tackles large-scale trajectory generation when real data are scarce and must generalize to new regions or cities. It introduces the Temporal Deaggregation Diffusion Model (TDDM), which separates “where people appear” from “how they move”: a spatial marginal prior guides a conditional diffusion model that reconstructs realistic time-evolving paths. Space is tiled into fixed-shape subregions and canonicalized via simple similarity transforms so one model can share parameters across different areas; generation becomes a mixture over regions, each conditioned on its local prior and then mapped back to global coordinates. A Transformer denoiser attends jointly to trajectory tokens, prior-map tokens, and diffusion-step tokens so spatial structure can inform every reverse step. Across standardized evaluations on Beijing (Geolife), Porto, and SF Bay (Cabspotting) with uniform preprocessing and diverse metrics, TDDM consistently improves distributional alignment and proportionality, matches or exceeds geometric realism, and shows stronger cross-region and cross-city robustness than GAN/VAE and diffusion baselines.

**Strengths:**

1. Original factorization of trajectory generation into a spatial prior (“where”) and temporal diffusion (“how”), avoiding sample-level conditioning.

2. Simple yet powerful region tiling with similarity transforms enables parameter sharing and cross-city generalization.

3. Thorough evaluation across three datasets with unified preprocessing, diverse metrics, and GAN/diffusion baselines.

4. Clear ablations isolating the impact of the spatial prior and region size trade-offs.

5. Practical significance for data-limited urban mobility, more controllable, scalable, and distribution-faithful synthesis.

**Weaknesses:**

1. Strong reliance on the quality of the spatial prior; robustness to sparse or biased priors is underexplored.

2. Preprocessing (1 Hz resampling + map-matching) may leak road constraints; limited analysis of training without map-matching.

3. Canonicalization via a single similarity transform can distort highly anisotropic geographies; no comparison to learned equivariance or multiscale warps.

4. Cross-city transfer still degrades path-length statistics; no targeted remedy (e.g., length regularization/domain adaptation).

5. Diffusion sampling cost and scaling to long trajectories or multi-agent settings are not addressed.

6. Privacy is acknowledged but lacks concrete DP mechanisms or re-identification audits.

**Questions:**

1. The preprocessing applies map-matching; could this leak road-network constraints into the training signal and inflate performance? Please report an ablation with (i) no map-matching and (ii) a weak-matching variant, and quantify each step’s contribution relative to the model itself.

2. To demonstrate that the model truly exploits the spatial prior H, could the authors provide attention heatmaps or attribution analyses showing interactions between H tokens and trajectory tokens?

3. How can the proposed method TDDM be applied to some concrete downstream tasks, like traffic assignment load, congestion proxy, routing robustness?

4. How does TDDM compare to a simpler pipeline, like an unconditional diffusion generator followed by post-hoc spatial reweighting to match the marginal prior H? This comparison would isolate the benefit of injecting the prior during denoising versus applying it after generation.

---

> ### Author Response · Authors · 2025-11-26
>
> We thank the reviewer for their detailed feedback and suggestions for additional experiments, which have strengthened the paper considerably.
>
> **W1:** We performed a spatial prior-dropout experiment (10–20% cell masking):
> | Metric        | 0% dropout              | 10% dropout       | 20% dropout   |
> | ------------- | ----------------- | ----------------- | ------------- |
> | TSTR (↓)      | **0.011 ± 0.006** | 0.026 ± 0.014     | 0.026 ± 0.014 |
> | KL(S ∥ R) (↓) | 0.301             | 0.224             | **0.223**     |
> | KL(R ∥ S) (↓) | **0.253**         | 0.277             | 0.277         |
> | KL_sym (↓)    | 0.277             | 0.251             | **0.250**     |
> | JS (↓)        | 0.059             | 0.052             | **0.051**     |
> | Density (↓)   | 0.019             | **0.013**         | **0.013**     |
> | Trip (↓)      | 0.031             | **0.027**         | 0.029         |
> | Length (↓)    | **0.004**         | 0.322             | 0.337         |
> | Pattern (↑)   | 0.917             | **0.933**         | 0.927         |
>
> Despite degraded input priors, TDDM retains strong performance across coverage ($KL(R||S)$), fidelity (TSTR), and structural metrics. Notably, $KL(R||S)$ reflects mode coverage and the ability to reproduce all real-world patterns. As it remains low even under dropout suggests that TDDM does not collapse or overly rely on any single region of the prior. This supports that the model leverages spatial priors, while **demonstrating robustness to sparse or biased priors**. We will include these results in the coming revision.
>
> **W2+Q1:** All baselines use the same preprocessed map-matched datasets, and evaluation is performed relative to this shared representation. GPS noise is reintroduced after map-matching, so map topology is not perfectly encoded in the training data.
>
> We conducted the requested map-matching ablation, retraining TDDM, DiffTraj, and Diffusion-TS on raw (non-map-matched) GPS data. Results averaged across three cities:
> | Metric | Diffusion-TS | DiffTraj | TDDM |
> |---|---|---|---|
> | TSTR ↓ | 0.016 ± 0.008 | 0.013 ± 0.007 | **0.011 ± 0.007** |
> | KL(S\|\|R) ↓ | 1.729 | 2.496 | **1.441** |
> | KL(R\|\|S) ↓ | **1.203** | 1.323 | 1.220 |
> | KL_sym ↓ | 1.466 | 1.910 | **1.330** |
> | JS ↓ | 0.243 | 0.300 | **0.220** |
> | Density ↓ | 0.044 | 0.080 | **0.033** |
> | Trip ↓ | 0.058 | 0.088 | **0.049** |
> | Length ↓ | 0.006 | 0.162 | **0.004** |
> | Pattern ↑ | 0.850 | 0.823 | **0.890** |
>
> Key findings:
> - TDDM maintains best overall performance without map-matching (8/9 metrics), demonstrating that improvements stem from the deaggregation framework, not the preprocessing.
> - Relative rankings are preserved across preprocessing conditions (TDDM: $KL_\text{sym}$ 0.277 → 1.330; Diffusion-TS: 1.153 → 1.466; DiffTraj: 1.232 → 1.910), confirming fair comparison.
> - Per-city breakdown shows consistent TDDM advantages on Geolife and Porto,
>
> **Question:** We ask the reviewer to clarify what “weak-matching variant” refers to.
>
> **W3:** Our local regions are square and isotropic by construction; similarity transforms do not distort them.
>
> **W4:** Differences in trip-length distributions between cities (e.g., Porto’s heavier tail vs SF’s concentrated distribution) cannot be inferred from the spatial prior alone. Since TDDM intentionally uses only spatial priors for zero-shot transfer, mismatches in length statistics reflect real city-specific temporal structure. Extending H with marginal length or time-of-day priors is a natural future direction, which we will note.
>
> **W5:** These extensions require rethinking tokenization and conditioning and are out of scope for this work. Our goal is to establish the spatial–temporal factorization; multi-agent and long-range modeling represent substantial next steps.
>
> **W6:** Privacy is outside the scope of this paper. We focus on improving the base generative quality. The improved fidelity and coverage we provide form a stronger foundation for future privacy-preserving variants.
>
> **Q2:** Our ablations (removing H, degrading H) directly quantify its impact: removing H degrades $KL_\text{sym}$ by 4x (Table 3), and dropout experiments show graceful degradation. These quantitative results are more reliable than attention maps for high-dimensional priors.
>
> **Q3:** TDDM can provide density maps, turning distributions, and repeated samples under fixed priors for load approximation, congestion proxies, and robustness analysis. For routing robustness, generating multiple plausible realizations under modified priors offers a straightforward way to explore how routing behavior might vary under perturbations or demand shifts. While not our focus, the factorization naturally supports such downstream uses. We will clarify this.

---

> > ### Author Response · Authors · 2025-11-26
> > **Part 2**
> >
> > **Q4:** We conducted the suggested experiment using rejection sampling on the unconditional model:
> >
> > | Metric | TDDM | 1x1 km | w/o spatial prior | w/o spatial prior + rejection |
> > |---|---|---|---|---|
> > | TSTR ↓ | **0.011 ± 0.006** | 0.024 ± 0.012 | **0.011 ± 0.006** | _0.014 ± 0.010_ |
> > | KL(S\|\|R) ↓ | **0.301** | _0.339_ | 1.569 | 1.925 |
> > | KL(R\|\|S) ↓ | **0.253** | _0.318_ | 1.098 | 1.252 |
> > | KL_sym ↓ | **0.277** | _0.328_ | 1.334 | 1.588 |
> > | JS ↓ | **0.059** | _0.071_ | 0.228 | 0.266 |
> > | Density ↓ | **0.019** | _0.022_ | 0.067 | 0.063 |
> > | Trip ↓ | **0.031** | _0.044_ | 0.074 | 0.081 |
> > | Length ↓ | **0.004** | 0.150 | 0.078 | _0.075_ |
> > | Pattern ↑ | 0.917 | **0.930** | _0.833_ | 0.860 |
> >
> > From the results we see that rejection sampling after unconditional generation worsens KL_sym from **1.334 → 1.588**, because it cannot create trajectories in under-covered regions. TDDM’s conditioning during denoising yields **4–5x** better distributional alignment (0.277 $KL_\text{sym}$), confirming the necessity of incorporating the prior in the diffusion process.

---

### Official Review · Reviewer_9NEn · 2025-11-01

**Soundness:** 2
**Presentation:** 1
**Contribution:** 2
**Rating:** 2
**Confidence:** 3

**Summary:**

This paper tackles the problem of trajectory generation by introducing TDDM, a hierarchical framework that leverages spatial priors to represent mobility and subsequently deaggregates them into trajectories. Experiments demonstrate the superiority of TDDM over existing baseline methods.

**Strengths:**

S1. This paper investigates the trajectory generation, which seems interesting and practical in real life.

S2. The paper proposes a hierarchical generative framework that separates spatial priors from temporal dynamics for large-scale trajectory generation.

S3. Experiments demonstrate that the proposed model achieves improvements in trajectory fidelity and spatial coverage compared with baselines.

**Weaknesses:**

W1. Novelty: While the paper proposes a hierarchical framework for trajectory generation by separating spatial priors and temporal dynamics, the overall model design is rather straightforward, and the novelty appears limited.

W2. Datasets: The experiments rely solely on taxi trajectory data, which restricts the generalizability of the findings. It is recommended to include additional domains to demonstrate robustness.

W3. Complexity Analysis: The paper lacks a discussion of the model’s time and space complexity.

W4. Efficiency Analysis: Reporting times comparison with baselines would strengthen the experimental evaluation.

**Questions:**

Q1: Some figures (e.g., Figure 2) are not referenced in the text, which affects.

Q2: Would it be more logical to place the overview part at the beginning of the methodology rather than at the end?

Q3: In Section 3, several equations are difficult to follow due to missing definitions—e.g., what does D represent?

Q4: Algorithms 1 and 2 are mentioned but not explained or discussed, making it hard for readers to understand their roles.

Q5: Please enlarge the text in Figure 3 to improve legibility.

Q6: Would it be better to move page 10 to the appendix, given the 9-page main body limit?

Q7: More details on the experimental setup (e.g., server) would enhance reproducibility.

Q8: Tables 1 and 2 are missing bottom borders—please fix for consistency.

Q9: There is only one subsection labeled “Section 3.1”. Should this be merged into Section 3, or are other subsections missing?

---

> ### Author Response · Authors · 2025-11-26
>
> We appreciate the reviewer's comments and address each point below.
>
> **W1:** While the architecture is intentionally simple, the *temporal deaggregation framework* is the core novelty: separating spatial priors from temporal dynamics enables zero-shot cross-city transfer—something not demonstrated by prior work. We refer to our detailed response to WVTu regarding methodological contributions.
>
> **W2:** Geolife is not a taxi dataset; it contains diverse transportation modes (walking, biking, bus, car, train) and activities (shopping, hiking, sightseeing). We will clarify this.
>
> **W3:** We include a detailed time/space complexity analysis: per-step cost $O((L+R)d^2 + (L+R)^2 d),$ with *fixed* sequence length $L$ and number of spatial prior patches $R$. Importantly, because we operate on fixed-size $3 \times 3$ km regions with 64 spatial tokens, per-trajectory complexity is constant across cities, and city-scale sampling grows only linearly in geographic coverage (city area).
>
> We will add this in the revision, even though baselines do not provide comparable analyses.
>
> **W4:** For TDDM's core sampling process, we measure approximately 340 ms per trajectory on an RTX 3090 Ti (see our response to Reviewer YR6q, Question 6, for detailed timing and memory analysis across batch sizes). As noted in W3, TDDM's per-trajectory cost remains constant regardless of city size, since we process fixed-size regions ($3 \times 3$ km) independently. This enables efficient scaling to arbitrary geographic coverage.
>
> **Q1+Q2+Q3+Q4+Q5+Q8+Q9:** We will address these formatting and presentation suggestions in the coming revision.
>
> **Q6:** The ICLR 2026 Author Guide (https://iclr.cc/Conferences/2026/AuthorGuide) explicitly excludes Ethics Statement, Reproducibility Statement, and LLM Usage sections from the 9-page main body limit. Page 10 content is compliant with submission guidelines.
>
> **Q7:** We have used systems with the following specifications:
>
> - CPU: AMD Ryzen 9 5900X
> - RAM: 128 GB
> - GPU: NVIDIA GeForce RTX 3090 Ti (24 GB GDDR6X)
>
> This will be added to the appendix in the coming revision.

---

### Official Review · Reviewer_WVTu · 2025-11-02

**Soundness:** 2
**Presentation:** 2
**Contribution:** 2
**Rating:** 2
**Confidence:** 5

**Summary:**

This paper introduces the Temporal Deaggregation Diffusion Model (TDDM), a hierarchical framework for generating large-scale, realistic trajectories. The core contribution is the factorization of the generation process: 1. The model first learns a **spatial prior**, $H$, which represents the marginal probability distribution of geographic occupancy. 2.  A conditional diffusion model then "deaggregates" this spatial prior, learning $p(x|H)$ to generate full temporal trajectories ($x$) that are consistent with the given spatial distribution.
To achieve out-of-distribution (OOD) generalization to new geographical regions, the authors propose a canonicalization technique. This method applies similarity transformations to normalize local spatial regions, allowing the diffusion model to learn spatio-temporal dynamics that are invariant to specific locations or orientations. The authors validate their method on a new benchmark of three large-scale datasets, demonstrating strong performance in both fidelity and OOD generalization compared to existing baselines.

**Strengths:**

1. The primary strength of this paper is its novel problem formulation. The factorization of trajectory generation into a spatial prior $p(H)$ and a conditional temporal dynamic $p(x|H)$ is an interesting idea. This deaggregation framework provides a natural mechanism for controllability (by manipulating $H$) and OOD generalization.

2. The paper directly tackles a major weakness of most trajectory generation models: being overfit to a specific city or road network.

3. The authors have constructed a new, large-scale benchmark from three diverse cities (Beijing, Porto, San Francisco) to test their claims. The quantitative results show a clear improvement over the chosen baselines, particularly in the challenging OOD "city-to-city" generalization experiments.

**Weaknesses:**

1.  The paper's claim to novelty in controllable generation and OOD generalization is weakened by the omission of several recent and highly relevant works. The current baselines (e.g., TimeGAN, DiffTraj) are not the state-of-the-art in controllable, topology-aware trajectory generation [1] [2]. The core idea of generating data to match a pre-defined utility (in this case, the spatial prior $H$) also shares conceptual similarities with prior work on utility-preserving generation, such as [3], which should be compared.

[1] Cola: Cross-city mobility transformer for human trajectory simulation, the Web Conferece 2024
[2] Controltraj: Controllable trajectory generation with topology-constrained diffusion model, KDD 24
[3] Generating mobility trajectories with retained data utility, KDD 2021

2. The OOD experiment, as designed, is insufficient to prove the effectiveness of the proposed techniques. The experiment involves fine-tuning on 25% of the target city's data. This is a very large amount of data for a "transfer" task. True OOD generalization should ideally be tested in a zero-shot setting, or at most a few-shot setting (e.g., 1-5% of data). Based on experience with human mobility generation, a standard generative model without any special transfer-learning design could likely achieve strong performance when fine-tuned on 20-25% of the data. Therefore, this experiment fails to demonstrate that the proposed canonicalization method offers any significant advantage over a simple, standard fine-tuning procedure. It cannot be used to validate the paper's central claims of OOD generalization.

3. A core premise of the paper that temporal dynamics (e.g., turning behavior, speed on a straight road) are universal and can be disentangled from the spatial context ($H$) via canonicalization, is a strong and largely unproven assumption. The experiments do not sufficiently rule out a simpler alternative hypothesis: that the model is simply learning a sophisticated "path-tracing" function. In this scenario, the OOD performance would be driven almost entirely by the quality of the provided target spatial prior $H$, rather than by the model successfully learning and transferring any truly "invariant" temporal dynamics.

4. The methodological novelty of the model architecture itself is limited. The core components, namely a Transformer backbone and a standard DDPM-style denoising loss, are a straightforward application of existing, well-established techniques in the field.

**Questions:**

See weaknesses

---

> ### Author Response · Authors · 2025-11-26
>
> We thank the reviewer for their feedback, particularly the path-tracing hypothesis, which prompted additional experiments that strengthen our claims. We address each point below.
>
> **W1:** We will add these works to Related Work and clarify distinctions. Empirical comparison is unfortunately infeasible:
>
> * **COLA** generates *discrete location IDs*, whereas TDDM generates *continuous GPS trajectories*. Metrics, representation, and model tasks differ fundamentally.
> * **ControlTraj** lacks critical preprocessing code for computing control attributes (GitHub [1] issues: #2, #4, #6, #9, #11). Running an incomplete pipeline would not yield a meaningful or fair comparison.
> * **TrajGen** omits both Location→Image and Image→Location modules, preventing end-to-end execution (GitHub [2] issues #1, #2).
>
> To maintain fairness, we focus our comparisons on reproducible and continuous-coordinate methods (Diffusion-TS, DiffTraj, TimeGAN, TimeVAE, COSCI-GAN) from top-tier venues (ICLR, NeurIPS, ICML).
>
> [1] https://github.com/Yasoz/ControlTraj
>
> [2] https://github.com/caochuntu/KDD2021_guizu
>
> **W2:** We would like to clarify a key misunderstanding: **we perform zero-shot generation, not fine-tuning**.
>
> * **Intra-city (25% → 100%)**: We train *from scratch* on a single fixed 25% geographic region (bottom-left quadrant; Figures 12–14), and evaluate on the remaining 75% *without any gradient updates or access to trajectories outside the training region*.
> * **City-to-city**: We train on one city and generate for a different city using only its spatial prior **H**—again with **no training** on the target.
>
> The model never sees target-region trajectories. Only the spatial prior H is provided at inference time. We will clarify this in the paper.
> Thus, both settings reflect **true zero-shot generalization**, enabled by canonicalization.
>
> **W3:** We appreciate this concern. To directly separate temporal dynamics from spatial guidance, we computed **speed KL divergence**, which captures *only* temporal behavior and cannot be inferred from H (H contains no timing or velocity information).
>
> **Speed KL - Intra-city Generalization (25% → 100%)**:
>
> | Model | Geolife   | Porto     | Cabspotting |
> | ----- | --------- | --------- | ----------- |
> | TDDM  | **0.011** | **0.019** | **0.007**   |
> | 25\%  | 0.046     | 0.027     | 0.230       |
>
> Speed degradation differs across regions (e.g., suburban vs. downtown SF), which would *not* occur if the model were merely tracing H. Porto's mild degradation (1.4x) suggests relatively uniform temporal structure.
>
> **Speed KL - City-to-City Generalization**:
>
> | Model       | Geolife   | Porto     | Cabspotting |
> | ----------- | --------- | --------- | ----------- |
> | Geolife     | **0.011** | 0.505     | 0.139       |
> | Porto       | 0.097     | **0.019** | 0.380       |
> | Cabspotting | 0.083     | 0.703     | **0.007**   |
>
> Two observations:
>
> 1. **If TDDM were simply path-tracing H, off-diagonal speed KL values would remain low.** Instead, they vary dramatically (0.083–0.703), revealing city-specific temporal patterns.
> 2. **Diagonal terms are consistently the lowest**, confirming the model reproduces city-specific dynamics.
>
> These results support that TDDM learns meaningful temporal behavior—not just tracing H.
>
> **W4:** We thank the reviewer for acknowledging the novelty of the formulation. While our backbone intentionally uses well-established components, our **methodological contribution lies in the factorization**:
>
> 1. **Temporal deaggregation with spatial priors** (Eq. 5), enabling controllable generation and zero-shot transfer.
> 2. **Canonicalization** via similarity transforms, providing spatial invariance *without* architectural constraints.
> 3. **Multi-modal tokenization** enabling trajectory–prior–diffusion interactions.
> 4. **Hierarchical mixture formulation** for region-wise generation at city-scale.
>
> This factorization yields **4x improvement in distributional metrics** over diffusion baselines (Table 1) and enables **zero-shot city-to-city generalization**—capabilities that prior methods with more complex architectures could not achieve. The simplicity of our design also ensures reproducibility and accessibility to practitioners.

---

> > ### Comment · Reviewer_WVTu · 2025-11-28
> >
> > Thank you for the authors’ response. However, my concerns remain largely unaddressed. I still believe that the technical contributions presented in this work are limited and do not meet the acceptance bar.
> >
> > In addition, after checking the datasets used in the experiments, I found that they are relatively small in scale: only 182 users for the human mobility dataset and 442/500 taxis for the taxi trajectory datasets. Such small-scale datasets cannot sufficiently serve as ground truth for evaluating large-scale trajectory generation or for demonstrating the scalability of the proposed method.
> >
> > Moreover, I have serious doubts about the reasonableness of transferring models between human mobility trajectories and taxi trajectories. These two types of trajectories exhibit fundamentally different characteristics. For example, taxi trajectories are driven by different passengers, show mixed-purpose travel, and do not follow consistent daily rhythms. In contrast, human mobility trajectories (e.g., GeoLife) reflect individual routine patterns that have higher predictability and stronger temporal regularity. Therefore, transferring between these highly heterogeneous datasets lacks clear justification.
> >
> > For future submissions, I recommend that the authors focus more on human mobility trajectory generation, where the individual-level temporal regularity provides a more meaningful and coherent setting. This direction is also better aligned with the stated motivation of modeling individual movement behavior.

---

> > > ### Author Response · Authors · 2025-12-03
> > >
> > > We thank the reviewer for their continued engagement.
> > >
> > > > However, my concerns remain largely unaddressed. I still believe that the technical contributions presented in this work are limited and do not meet the acceptance bar.
> > >
> > > We sincerely believe we have substantively responded to the concerns raised in the original review.
> > >
> > > > In addition, after checking the datasets used in the experiments, I found that they are relatively small in scale: only 182 users for the human mobility dataset and 442/500 taxis for the taxi trajectory datasets. Such small-scale datasets cannot sufficiently serve as ground truth for evaluating large-scale trajectory generation or for demonstrating the scalability of the proposed method.
> > >
> > > **Dataset scale**: We respectfully correct a framing issue. While Geolife contains 182 users, it comprises **17,621 trajectories**. Porto and Cabspotting cover taxi fleets for a full year and 30 days of continuous collection respectively, yielding hundreds of thousands of trajectory segments after preprocessing. These scales match or exceed those used in prior trajectory generation work.
> > >
> > > We are unaware of larger publicly available datasets. By using three public datasets we meet or exceed standard practice:
> > > - TrajGen (KDD 21): Singapore (private)
> > > - DiffTraj (NeurIPS 23): Chengdu (private), Xi'an (private), Porto (public)
> > > - ControlTraj (KDD 24): Chengdu (private), Xi'an (private), Porto (public)
> > >
> > > > Moreover, I have serious doubts about the reasonableness of transferring models between human mobility trajectories and taxi trajectories. [...]
> > >
> > > **Cross-dataset transfer**: Our contribution is **geographic generalization** (Beijing→Porto→San Francisco), not behavioral transfer. The city-to-city experiments (Table 9) demonstrate zero-shot generation in new cities using only spatial priors—a capability no baseline achieves. Dataset heterogeneity actually strengthens this evaluation: robust cross-city performance across multiple complementary metrics—TSTR (0.009-0.017, matching in-distribution performance), symmetric KL (0.263-0.982), and Trip Error (0.022-0.047)—validates that canonicalization captures transferable motion patterns despite domain differences.

---

### Author Response · Authors · 2025-11-26
**General rebuttal (to all reviewers and AC)**

We thank all reviewers for their constructive feedback. We address concerns regarding novelty, baselines, and generalization below. Importantly:

**(1) All OOD experiments are zero-shot.**
The model is *never* fine-tuned on target-city trajectories in the generalization experiments. Intra-city: trained on 25% of a city, evaluated on the remaining 75% with *no* target-region gradients. City-to-city: trained on City A, evaluated on City B using only its spatial prior.

**(2) Our contribution is a *factorization*, not an architectural tweak.**
The temporal deaggregation framework—canonicalization + spatial priors + mixture-based region sampling—is what enables zero-shot cross-region generation. The Transformer+DDPM backbone is intentionally simple and reproducible.

**(3) Additional experiments requested by reviewers:** Speed KL, spatial prior dropout, realism with map-matching, inference cost, and introducing a post-hoc reweighting baseline. All support the correctness and robustness of TDDM.

**(4) Missing baselines (COLA [1], ControlTraj [2], TrajGen [3]).** COLA solves a fundamentally different problem, making comparison infeasible. ControlTraj and TrajGen cannot be reproduced fairly, as key components are absent in the public repositories. We therefore focus on reproducible continuous-coordinate baselines (Diffusion-TS, DiffTraj, TimeGAN, TimeVAE, COSCI-GAN).

**(5) Temporal dynamics are learned, not path-traced.** Speed KL (a purely temporal metric) cannot be inferred from the spatial prior. TDDM achieves the best results on 2/3 datasets and shows dataset-specific degradation under cross-city transfer—both inconsistent with path-tracing. This is futher corroborated by the dropout experiment.

We provide detailed replies below.

[1] Wang, Yu, et al. "Cola: Cross-city mobility transformer for human trajectory simulation." Proceedings of the ACM Web Conference 2024.

[2] Zhu, Yuanshao, et al. "Controltraj: Controllable trajectory generation with topology-constrained diffusion model." Proceedings of the 30th ACM SIGKDD Conference on Knowledge Discovery and Data Mining. 2024.

[3] Cao, Chu, and Mo Li. "Generating mobility trajectories with retained data utility." Proceedings of the 27th ACM SIGKDD conference on knowledge discovery & data mining. 2021.

---

### Author Response · Authors · 2025-12-03

Dear Area Chair,

We have made substantial revisions to our paper in response to reviewer feedback during the rebuttal period. Below is a comprehensive list of all changes, organized by category, with references to the specific reviewer concerns they address. We believe these revisions substantially strengthen the paper and address all key concerns raised by all reviewers.

**1. Related Work & Methodology Clarifications**

- Added COLA, ControlTraj, and TrajGen to the Introduction and Related Work (Appendix A) and clarified distinction from TDDM's approach (WVTu W1)
- Reorganized Related Work section for improved flow and clarity
- Added explanation of Algorithms 1 and 2 before presenting them in Section 3, describing training procedure (random region sampling, spatial prior computation) and sampling procedure (zero-shot transfer via spatial priors alone) (9NEn Q4)
- Clarified zero-shot generalization setup in Section 4.3 (WVTu W2)
- Clarified Geolife dataset contains diverse transportation modes, not just taxi data in Dataset and Preprocessing paragraph in Section 4 (9NEn W2)
- Clarified training region selection with references to Figures 12-14 (YR6q Q1)
- Additional minor clarifications addressing 9NEn Q3, Q9

**2. New Experimental Results**

Speed KL Divergence:
- Added Speed KL for measuring temporal fidelity via speed distribution divergence to all experimental tables: unconditional generation, ablation study, 25% generalization, city-to-city transfer, map dropout robustness, and map matching evaluation (WVTu W3, YR6q Q5)

New Figures and Tables:
- Added Map Dropout Robustness Table (0%, 10%, 20% dropout) demonstrating robustness to sparse/biased priors, Table 10 in Appendix (L1Qa W1)
- Added Map Matching Ablation Table comparing TDDM, DiffTraj, and Diffusion-TS without map matching preprocessing, Table 9 in Appendix (L1Qa W2, L1Qa Q1)
- Added Rejection Sampling Baseline comparison isolating benefit of conditioning during denoising, Table 2 (L1Qa Q4)
- Added trajectory length distribution histogram figures comparing cities (e.g., Porto vs Cabspotting) to explain cross-city length error differences, Figures 16-21 (YR6q Q4)

**3. New Appendix Section: Computational Complexity and Runtime Analysis**

- Added new appendix section with theoretical complexity analysis, empirical runtime and memory measurements, and hardware specifications, Appendix D. (9NEn W3, W4, Q7, YR6q Q6)

**4. Analysis & Discussion Improvements**

- Added clarification in Section 4.3, paragraph 2, noting that per-city analysis reveals a more substantial variation, with reference to full breakdowns in appendix (YR6q W1)
- Updated phrasing around 25% generalization results for clarity (YR6q W1)

**5. Future Work Additions**

- Added discussion of extending spatial prior representation H with additional structured information, including: marginal length or time-of-day priors, directional priors for traffic handedness, road hierarchy information, and temporal marginals for improved cross-city temporal fidelity (L1Qa W4, YR6q Q7, Q8, Q9)

**6. Formatting & Presentation**

- Added underline formatting for second-best results in all tables
- Remove bottom borders for consistency in Tables 1 & 2 (9NEn Q8)

We thank the reviewers for their constructive feedback, which has led to a more comprehensive evaluation of our method.

---

### Meta-Review · Area_Chair_uMqx · 2026-01-04

**Summary:**

This paper introduces the Temporal Deaggregation Diffusion Model (TDDM), a hierarchical framework that first represents mobility using spatial priors and then deaggregates them into trajectories. This method supports cross-city transfer learning and uses Beijing (GeoLife), Porto, and San Francisco as the evaluation benchmarks.

This paper addresses a significant research problem, but most reviewers are rather negative towards this paper, even after rebuttal. The common concerns can be summarized as follows: (1) the methodological novelty is limited; (2) the rationale behind the proposed approach is not sufficiently clear, particularly regarding why city-to-city transfer is expected to succeed; (3) the core assumptions—such as the claim that temporal dynamics can be disentangled from spatial context—are not adequately validated; and (4) the robustness of the method is not fully demonstrated. In addition, multiple reviewers report persistent issues with presentation.

After carefully reviewing the rebuttal and the updated manuscript, I believe that the work would still benefit from substantial revision based on the reviewers’ comments. Therefore, I recommend rejection.

**Reviewer Concerns:**

Some presentation issues have been addressed by the rebuttal, and some experiments were appended.

However, the rationale behind the methodology and validations on some key assumptions are still missing.

**Reviewer Scores:**

Reviewer WVTu 2->2, will not change the score.

Reviewer 9NEn 2->2, will not change the score.

Reviewer L1Qa 4->4, will not change the score.

Reviewer YR6q 6->6, will not change the score.

---

### Decision · Program_Chairs · 2026-01-26

Reject